EMBO
Molecular Medicine

# *SLC25A46* is required for mitochondrial lipid homeostasis and cristae maintenance and is responsible for Leigh syndrome

Alexandre Janer[1,2], Julien Prudent[2,3,†], Vincent Paupe[1,2,†], Somayyeh Fahiminiya[1], Jacek Majewski[1], Nicolas Sgarioto[2], Christine Des Rosiers[4,5], Anik Forest[4,5], Zhen-Yuan Lin[6], Anne-Claude Gingras[6,7], Grant Mitchell[8], Heidi M McBride[2,3] & Eric A Shoubridge[1,2,*]

## Abstract

Mitochondria form a dynamic network that responds to physiological signals and metabolic stresses by altering the balance between fusion and fission. Mitochondrial fusion is orchestrated by conserved GTPases MFN1/2 and OPA1, a process coordinated in yeast by Ugo1, a mitochondrial metabolite carrier family protein. We uncovered a homozygous missense mutation in *SLC25A46*, the mammalian orthologue of *Ugo1*, in a subject with Leigh syndrome. SLC25A46 is an integral outer membrane protein that interacts with MFN2, OPA1, and the mitochondrial contact site and cristae organizing system (MICOS) complex. The subject mutation destabilizes the protein, leading to mitochondrial hyperfusion, alterations in endoplasmic reticulum (ER) morphology, impaired cellular respiration, and premature cellular senescence. The MICOS complex is disrupted in subject fibroblasts, resulting in strikingly abnormal mitochondrial architecture, with markedly shortened cristae. SLC25A46 also interacts with the ER membrane protein complex EMC, and phospholipid composition is altered in subject mitochondria. These results show that SLC25A46 plays a role in a mitochondrial/ER pathway that facilitates lipid transfer, and link altered mitochondrial dynamics to early-onset neurodegenerative disease and cell fate decisions.

**Keywords** Leigh syndrome; mitochondrial architecture; phospholipid transfer; SLC25A46

**Subject Categories** Genetics, Gene Therapy & Genetic Disease; Metabolism; Neuroscience

## Introduction

Mitochondria are dynamic organelles that change their morphology in response to physiological signals and metabolic stresses. This is achieved by altering the balance between fusion and fission, and by changes in the architecture of the organelle itself, in particular the invaginations of the inner mitochondrial membrane, the cristae, that house the protein complexes of the oxidative phosphorylation (OXPHOS) system. In mammals mitochondrial fusion is mediated by dynamin-related GTPases on the outer membrane (MFN1, MFN2) and by OPA1 on the inner membrane. Fusion of the mitochondrial network occurs in response to stress (Tondera *et al*, 2009; Shutt *et al*, 2012), in conditions of nutrient deprivation (Gomes *et al*, 2011; Rambold *et al*, 2011), and during the G1/S phase of the cell cycle (Mitra *et al*, 2009). Dominantly inherited neurological phenotypes are caused by mutations in profusion proteins: MFN2 mutations are associated with Charcot–Marie–Tooth type 2A (Zuchner *et al*, 2004) and mutations in OPA1 are the major cause of dominantly inherited optic atrophy (Alexander *et al*, 2000; Delettre *et al*, 2000). OPA1 has an additional role in the maintenance of cristae morphology and its remodeling in response to metabolic signals (Frezza *et al*, 2006; Meeusen *et al*, 2006). For example, apoptotic stimuli promote cristae opening, allowing release of cytochrome *c* (Scorrano *et al*, 2002; Frezza *et al*, 2006), nutrient availability modulates cristae number and structure (Patten *et al*, 2014; Sood *et al*, 2014), cristae shape modulates respiration through the assembly of respiratory chain super complexes (Cogliati *et al*, 2013), and stabilization of cristae, by mild overexpression of OPA1, can protect against ischemic insults and muscle atrophy (Civiletto *et al*, 2015; Varanita *et al*, 2015). The cleaved form of Opa1 is not required for fusion, but contributes to mitochondrial division, indicating a highly

1 Department of Human Genetics, McGill University, Montreal, QC, Canada
2 Montreal Neurological Institute, McGill University, Montreal, QC, Canada
3 Department of Neurology and Neurosurgery, McGill University, Montreal, QC, Canada
4 Department of Nutrition, Université de Montréal, Montreal, QC, Canada
5 Research Centre, Montreal Heart Institute, Montreal, QC, Canada
6 Lunenfeld-Tanenbaum Research Institute, Mount Sinai Hospital, Toronto, ON, Canada
7 Department of Molecular Genetics, University of Toronto, Toronto, ON, Canada
8 Division of Medical Genetics, Department of Pediatrics, CHU Sainte-Justine and Université de Montréal, Montreal, QC, Canada
*Corresponding author. Tel: +1 514 398 1997; Fax: +1 514 398 1509; E-mail: eric@ericpc.mni.mcgill.ca
†Co-second authors

regulated, dual role for this GTPase in fusion and fission (Anand *et al*, 2014). These studies demonstrate that dynamic transitions in mitochondrial architecture are essential to tune mitochondrial function to the changing needs of the cell.

In yeast, mitochondrial fusion requires a third player, Ugo1, which associates with the homologues of MFN2 (Fzo1 in yeast) and OPA1 (Mgm1 in yeast) (Sesaki & Jensen, 2004). Ugo1, a degenerate member of the mitochondrial metabolite carrier family, is a multi-span outer membrane protein that assembles independent of the conventional protein import machinery (Vogtle *et al*, 2015), and is thought to coordinate fusion of the outer and inner mitochondrial membranes, perhaps playing a role in lipid mixing downstream of mitochondrial tethering (Hoppins *et al*, 2009). Ugo1 also appears to play an important additional role in the maintenance of cristae architecture, as mitochondria in *Ugo1* deletion strains are practically devoid of cristae and have very few contact sites between the inner and outer mitochondrial membranes (Harner *et al*, 2011).

Mitochondrial contact site and cristae organizing system (MICOS) is a recently discovered complex, consisting of at least seven proteins in a complex of about 700 kDa, that is indispensable for the maintenance of mitochondrial cristae junctions (Harner *et al*, 2011; Hoppins *et al*, 2011; van der Laan *et al*, 2012; Pfanner *et al*, 2014; Guarani *et al*, 2015). An evolutionary analysis of MICOS showed a widespread distribution of the central MICOS proteins among eukaryotes, and loss of the complex in organisms with simplified cristae, underscoring the ancient origins and importance of MICOS in cristae formation (Munoz-Gomez *et al*, 2015; Huynen *et al*, 2016). Disruption of this complex results in detachment of membrane contact sites, producing parallel stacks of cristae that have lost the connection between the inner boundary membrane and the outer mitochondrial membrane (Harner *et al*, 2011; Hoppins *et al*, 2011; van der Laan *et al*, 2012). Thus, both MICOS, OPA1, and Ugo1 have important roles in the maintenance of cristae architecture, but how they interact is not yet clear.

Here, we show that a member of the mitochondrial metabolite carrier family, SLC25A46, that we found mutated in Leigh syndrome, an early-onset neurodegenerative disease, is the likely orthologue of Ugo1. A recent study reported mutations in SLC25A46 in patients with optic atrophy, axonal Charcot–Marie–Tooth disease (CMT), and cerebellar atrophy (Abrams *et al*, 2015), and in a patient with an optic atrophy spectrum disorder (Nguyen *et al*, 2016), but neither study elucidated its molecular function. We show that SLC25A46 functions upstream of the MICOS complex and is required for the maintenance of mitochondrial cristae architecture. We further identified interactions with the ER/mitochondrial contact site complex (EMC) that has been implicated in lipid transport to mitochondria (Lahiri *et al*, 2014) and show that loss of SLC25A46 function produces alterations in mitochondrial phospholipid composition, providing insight into the molecular function of SLC25A46.

## Results

### A mutation in *SLC25A46* is responsible for Leigh syndrome and impairs respiration

We investigated a girl born to first cousin French Canadian parents. Clinically, the subject presented with Leigh syndrome, an early-onset and fatal neurodegenerative disease associated with bilaterally symmetric lesions in the brainstem, basal ganglia, and spinal cord (Leigh, 1951). Additional clinical details are reported in the Appendix.

Several candidate genes failed to reveal a causative mutation, so we used whole-exome sequencing to search for a homozygous mutation. This analysis identified a homozygous missense mutation (c.425C > T) in *SLC25A46,* a gene coding for a member of the mitochondrial metabolite carrier family. Sanger sequencing of cDNA and gDNA isolated from subject fibroblasts confirmed the presence of the mutation (Fig 1A). The SLC25 family of mitochondrial metabolite carriers comprises more than 50 proteins in humans (Monne & Palmieri, 2014). Of the approximately half that have been functionally characterized, the majority transport specific metabolites across the inner mitochondrial membrane. The metabolite carrier proteins have a characteristic tripartite structure consisting of repeats of ~100 amino acids, each with two transmembrane alpha helices (Palmieri, 2014). The mature carrier protein thus consists of six transmembrane helices that form around an aqueous pore through which small metabolites are transported. Figure 1B shows a schematic representation of SLC25A46 and the mutation predicted to cause an amino acid substitution (p.Thr142Ile) between helix one and two. Phylogenetic analysis of the mitochondrial metabolite carrier proteins in human, yeast, and plants showed that SLC25A46 is most closely related to Ugo1, an outer membrane protein in yeast that is essential for mitochondrial membrane fusion (Sesaki & Jensen, 2001; Coonrod *et al*, 2007; Hoppins *et al*, 2009; Anton *et al*, 2011; Papic *et al*, 2011), suggesting that they are orthologous (Haferkamp & Schmitz-Esser, 2012). Mitochondrial metabolite carriers have a highly conserved consensus sequence, PX(D,E)XX(K,R), at the C-terminal end of the odd numbered transmembrane alpha helices in all three repeats, the charged residues forming salt bridges that close the pore on the matrix side (Monne *et al*, 2013). These sequences are not conserved in SLC25A46 (Fig EV1), suggesting that, while it is a member of this protein family, it does not likely have a conventional metabolite carrier function. The N-terminus of SLC25A46 is also exceptionally long (~100 amino acids) compared to most other mitochondrial metabolite carriers (< 20 amino acids).

The p.Thr142Ile SLC25A46 variant was virtually undetectable in subject fibroblasts by immunoblot analysis (Fig 1D), suggesting that the missense mutation destabilizes the protein, perhaps by preventing its insertion into the membrane. BN-PAGE analysis of the oxidative phosphorylation (OXPHOS) complexes consistently showed only a specific assembly defect in complex IV in subject fibroblasts (Fig 1C), which correlated with decreased steady-state levels of two complex IV subunits, COX2 and COX4 (Fig 1D), a hallmark of such assembly defects (Zhu *et al*, 1998). Retroviral expression of the wild-type HA-tagged SLC25A46 cDNA completely rescued complex IV assembly (Fig 1C and D), confirming that it is the causative gene. To investigate whether the OXPHOS assembly defect resulted in a significant effect on overall cellular respiration, we measured the oxygen consumption rate in subject fibroblasts on a Seahorse apparatus (Fig 1E). This showed an approximately 50% decrease in basal oxygen consumption, with no measureable change in maximal oxygen consumption, which could be partially rescued by retroviral expression of *SLC25A46-GFP* (Fig 1E, top panel). siRNA-mediated suppression of SLC25A46 in control fibroblasts phenocopied the basal oxygen consumption defect in the subject cells (Fig 1E,

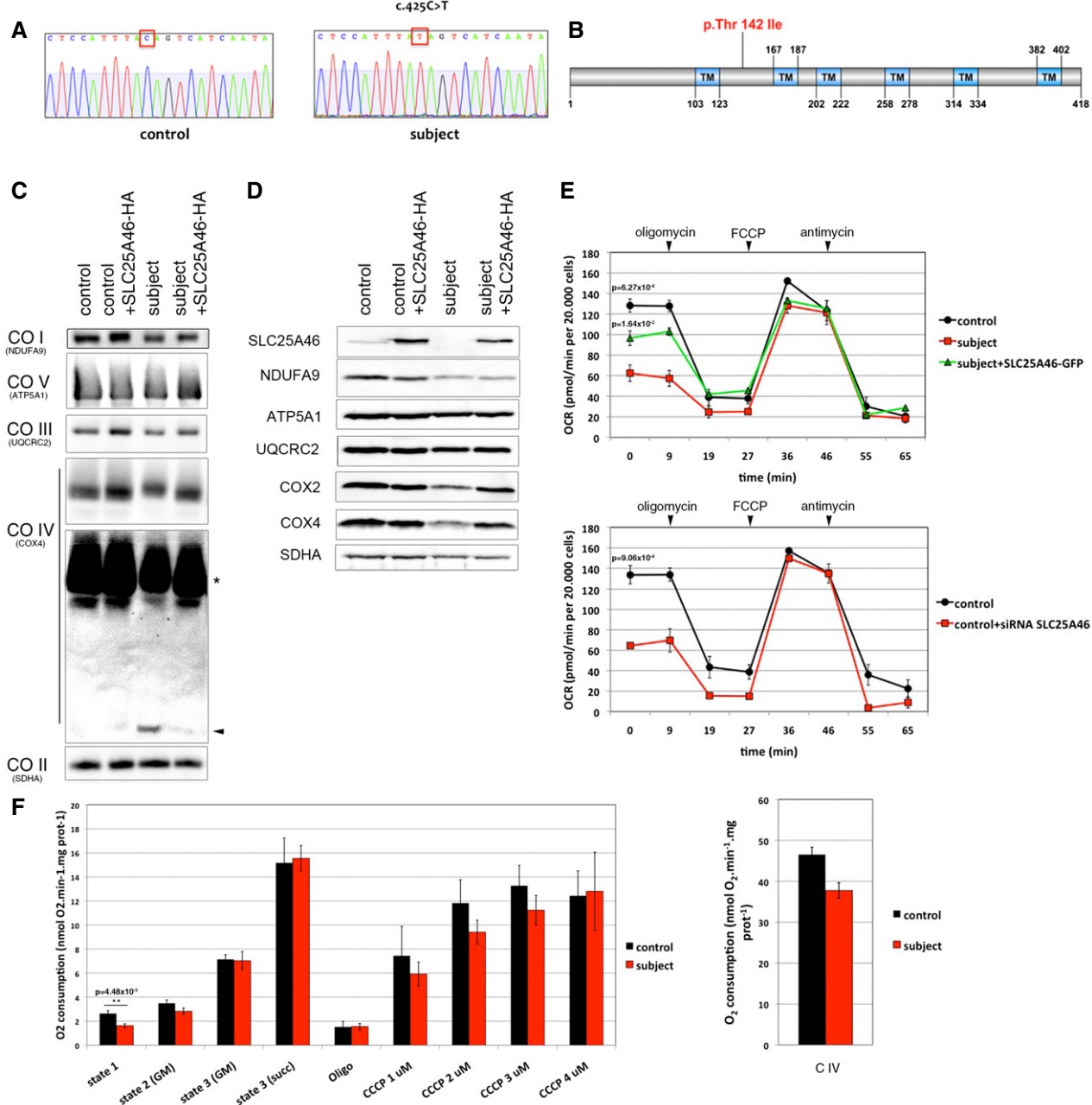

**Figure 1.  Loss of SLC25A46 function impairs respiration.**

A   DNA sequence analysis of *SLC25A46* cDNA showing a homozygous c.425C > T mutation in the subject fibroblasts compared to the control.

B   Schematic representation of SLC25A46 protein and position of the mutated p.Thr142Ile amino acid.

C   BN-PAGE analyses of OXPHOS mitochondrial complexes in fibroblasts from control and subject, overexpressing an HA-tagged version of wild-type *SLC25A46*. Each of the five OXPHOS complexes (I–V) was visualized with a subunit-specific antibody that recognizes the native complex as follows: COI (NDUFA9), COII (SDHA), COIII (UQCRC1), COIV (COX4), COV (ATP5A1). Overexposure of the blot shows a defect in assembled complex IV (asterisk) and some free unassembled COX4 subunit in the subject (arrowhead). Complex II was used as a loading control.

D   Immunoblot analysis of the same samples as in (C) for expression of SLC25A46 and individual structural subunits of the five OXPHOS complexes. The 70 kDa subunit of complex II (SDHA) was used as a loading control.

E   Seahorse analysis of oxygen consumption rate in fibroblasts from control compared to subject fibroblasts with or without expression of an EGFP-tagged version of wild-type *SLC25A46* (top panel) and in fibroblasts from control compared to those in which SLC25A46 is suppressed by a specific siRNA (bottom panel). $n = 3$, data represent mean $\pm$ SEM and *P*-values were calculated using a *t*-test.

F   Polarographic analysis of oxygen consumption in permeabilized fibroblasts from control and subject. Left panel: State 1: no exogenous substrates; state 2: addition of glutamate and malate (GM); state 3 (GM): addition of ADP; state 3 (Succ): addition of succinate; Oligo: inhibition of ATP synthase by oligomycin. Serial addition of 1–4 μM CCCP measuring maximal uncoupled respiration (control $n = 4$ vs. subject $n = 5$). Right panel: Complex IV-driven $O_2$ consumption was measured in the presence of antimycin A, TMPD and ascorbate (control $n = 4$ vs. subject $n = 5$). Data represent mean $\pm$ SEM and *P*-values were calculated using a *t*-test.

Source data are available online for this figure.

bottom panel), confirming the specificity of the respiration defect. To further probe the cellular respiration phenotype, we carried out additional oxygen consumption measurements using a conventional Clark-type electrode on digitonin-permeabilized fibroblasts, in which it is possible to interrogate the activity of the components of the OXPHOS system in the presence of different substrates (Fig 1F). These results confirmed that basal oxygen consumption was reduced in the subject and that maximal uncoupled electron flow was no different in the subject vs. control (Fig 1F, left panel). Significantly, there was no difference in state 3 respiration with either malate/glutamate or succinate in the subject vs. control, demonstrating that this part of the respiratory chain is intact. Using TMPD/ascorbate to measure the maximum flux through complex IV, we observed a small decrease in the subject cells, consistent with the assembly defect, but which was, however, not statistically significant (Fig 1F, right panel).

## SLC25A46 localizes to the outer mitochondrial membrane and loss of function causes mitochondrial hyperfusion

To confirm that SLC25A46 was an integral membrane protein, we assessed whether it could be extracted by alkaline carbonate (Fig 2A). As expected, SLC25A46 remained in the pellet fraction as did the control membrane protein MFN2. To determine whether the protein localized to the inner or outer membrane, we assessed the sensitivity of SLC25A46 to proteinase K digestion, using as control markers of all mitochondrial sub-compartments (Fig 2B). SLC25A46 behaved like MFN2, strongly suggesting that it, like Ugo1 in yeast, localizes to the outer mitochondrial membrane. Similar results were reported in Abrams et al (2015). The only other members of the mammalian mitochondrial carrier family reported to localize to the mitochondrial outer membrane are MTCH1 (SLC25A49), a protein that interacts with presenilin (Lamarca et al, 2007), and MTCH2 (SLC25A50), which has been co-opted by the apoptosis machinery, acting on the outer mitochondrial membrane as a receptor for tBid to recruit the pro-apoptotic protein BAX of the Bcl-2 family (Robinson et al, 2012). Neither MTCH1 or MTCH2 possess the conserved carrier domains typical of the SLC25 family (Fig EV1).

Considering the evolutionary relationship with the yeast fusion regulator Ugo1, we examined mitochondrial morphology in subject and control fibroblasts following siRNA-mediated suppression of SLC25A46 (Fig 2D). Fibroblasts from the subject, and from the siRNA-treated control fibroblasts, showed a striking hyperfused phenotype (quantified in Fig 2E), opposite of what is found in yeast Ugo1 loss-of-function mutants (Coonrod et al, 2007; Hoppins et al, 2009). Expression of SLC25A46-GFP rescued this phenotype in subject fibroblasts. The mitochondrial hyperfusion seems much more pronounced in fibroblasts than that reported in HeLa cells (Abrams et al, 2015), and interestingly, increased mitochondrial fission was reported in fibroblasts from the patient with optic atrophy spectrum disorder with a homozygous frameshifting mutation (Nguyen et al, 2016).

We next examined the effects of these manipulations on key components of the mitochondrial fission/fusion apparatus. Immunoblot analysis (Fig 2C) of the subject fibroblasts showed an accumulation of the profission protein DRP1 and the short OPA1 isoforms, which have been reported to trigger mitochondrial fragmentation (Anand et al, 2014). However no change was observed in the level of the fission factor MFF compared to controls. In subject fibroblasts, and in cells suppressed for SLC25A46, the level of MFN2 was insensitive to SLC25A46 loss or its overexpression. These results suggest that the hyperfusion phenotype cannot be explained by changes in profission or fusion factors.

DRP1 recruitment to the outer mitochondrial membrane at ER/mitochondrial contacts and its oligomerization are key steps in mitochondrial scission (Palmer et al, 2011). To investigate whether the striking hyperfused phenotype caused by lack of SLC25A46 was linked to a defect in DRP1 recruitment to mitochondria, we carried out a detailed analysis of DRP1 localization by immunofluorescence and subcellular fractionation, and in addition investigated its oligomerization state (Fig 3). Immunofluorescence analysis showed that DRP1 was recruited to mitochondria in subject cells (Fig 3A). The quantification of the overlapping signal between DRP1 and mitochondria (TOMM20) using Manders and Pearson coefficients (Fig 3B, top panel), as well as the calculation of the proportion of DRP1 present in mitochondria or cytosol (Fig 3B, bottom panel), demonstrated that DRP1 is only slightly less present at mitochondria in subject cells compared to control cells. On the other hand, the DRP1 content in the cytoplasm of subject cells was increased relative to controls. These results were entirely corroborated by subcellular fractionation and quantitative immunoblot analysis (Fig 3C), which showed a 1.85 increase in total cellular DRP1 and a small

**Figure 2. SLC25A46 controls mitochondrial dynamics.**

A   Alkaline carbonate extraction of mitochondria from control fibroblasts. Immunoblot analysis shows that SLC25A46 is an integral membrane protein. SDHA (soluble, membrane-associated protein) and MFN2 (integral inner membrane protein) were used as controls.

B   Proteinase K digestion assay on mitochondria from control fibroblasts. Mitochondria were exposed to an increasing concentration of proteinase K to determine the submitochondrial localization of SLC25A46. SLC25A46 behaves as an outer membrane protein. MFN2 was used as a control for outer membrane protein, AIFM1 for protein present in the inter-membrane space, HCCS and SCO1 for inner membrane proteins, and MRPL14 for matrix protein.

C   Immunoblot analysis of mitochondria from the same samples as in (D), for expression of SLC25A46 and known molecular components of the fusion/fission machinery. SDHA was used as a loading control.

D   Immunofluorescence analysis of the mitochondrial network. Control and subject fibroblasts, overexpressing an EGFP-tagged version of the wild-type SLC25A46 protein (green) and control fibroblasts treated with an siRNA against SLC25A46, were stained with the mitochondrial marker TOMM20 (red) and the nuclear marker DAPI (blue). Scale bars: 10 μm.

E   Quantification of the effect of SLC25A46 on the mitochondrial network shape. In each experimental condition described in (D), 150 cells were analyzed and the mitochondrial network organization was classified as intermediate, fused, or fragmented, in three independent experiments. Data represent mean $\pm$ SEM and P-values were calculated using a t-test: control vs. patient, $P = 1.10 \times 10^{-4}$; control vs. siRNA SLC25A46, $P = 1.16 \times 10^{-4}$; subject vs. subject + SLC25A46-GFP, $P = 2.06 \times 10^{-5}$.

Source data are available online for this figure.

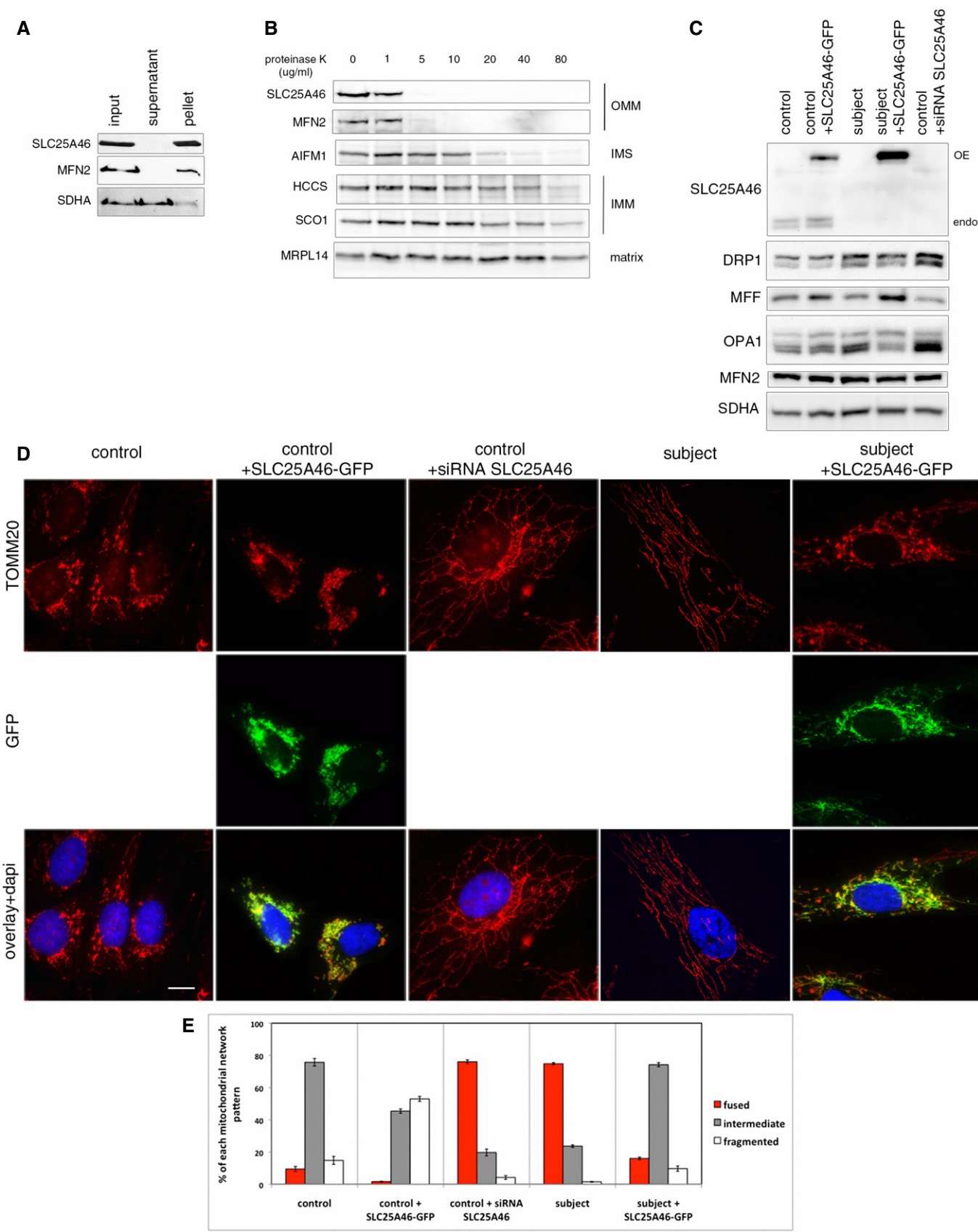

Figure 2.

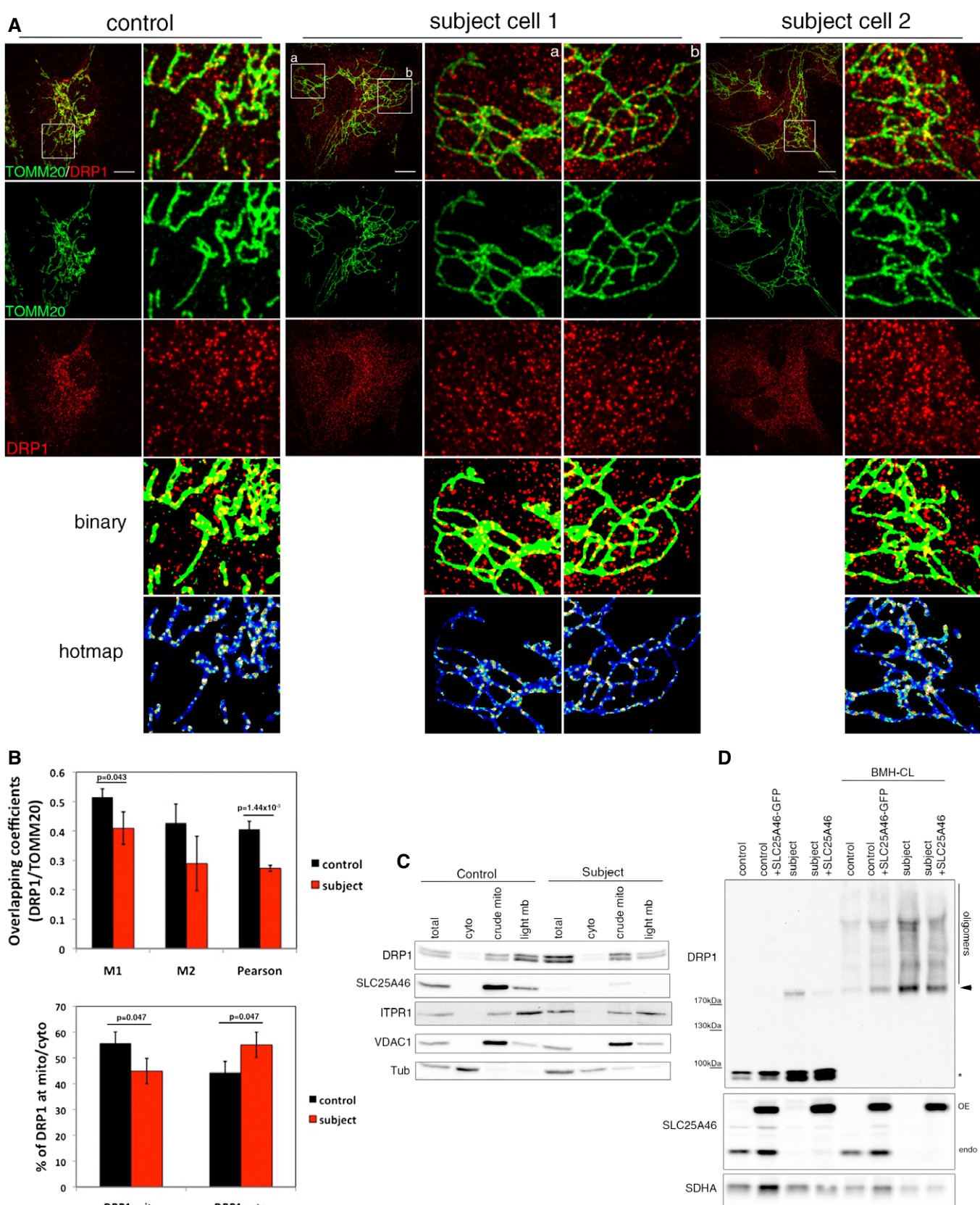

Figure 3.

◄

**Figure 3.  Role of SLC25A46 in mitochondrial DRP1 recruitment.**

A   Immunofluorescence analysis of DRP1 localization in control and subject fibroblasts. Mitochondria were visualized with TOMM20 (green) and endogenous DRP1 in red. Binary indicates thresholded binarized images and the hot map shows thresholded and binarized images from DRP1 and mitochondria combined to create an image of mitochondrial DRP1. The mitochondrial DRP1 created image was then merged with the mitochondrial image. False colors show the accumulation of DRP1 at mitochondria. Scale bars: 10 μm.

B   Quantitation of the DRP1 localization by immunofluorescence analysis. Upper panel: Co-localization of TOMM20 and DRP1 in control and subject fibroblasts was calculated by Mander's and Pearson's coefficients. Bottom panel: Quantification of DRP1 localization in the cytosol and the mitochondria. At least 15 cells per condition were analyzed in three independent experiments. Data represent mean ± SD and *P*-values were calculated using a Student's *t*-test.

C   Analysis of DRP1 localization in control and subject fibroblasts by immunoblot analysis after cellular fractionation. ITPR1 was used as an ER marker, VDAC1 as a mitochondrial marker, and alpha-tubulin as a cytosolic marker.

D   Immunoblot analysis of mitochondrial DRP1 oligomers in control and subject fibroblasts, overexpressing an EGFP-tagged version of wild-type SLC25A46. Crude mitochondria were treated or not with the BMH cross-linker. SDHA was used as a loading control. Monomers (asterisk), dimers (arrowhead), and oligomers of DRP1 are indicated.

Source data are available online for this figure.

decrease in the ratio cytosolic/mitochondrial DRP1 (0.2–0.3) in the subject vs. control. Interestingly, even though there is a slight decrease in mitochondrial DRP1 recruitment in the subject, analysis of its oligomerization state through cross-linking with BMH demonstrated an increase in DRP1 oligomers in subject vs. control cells (Fig 3D, *: monomer, arrowhead: dimer). The phosphorylation status of DRP1 on serines 616 or 637, which has been shown to be involved in mitochondrial dynamics, was also unaltered in subject cells compared to controls (Fig EV2). We conclude that an inability to recruit or oligomerize DRP1 at the mitochondrial membrane does not account for the mitochondrial hyperfusion phenotype in subject cells.

## Loss of SLC25A46 results in loss of mitochondrial cristae and destabilization of the MICOS complex

We next performed transmission electron microscopy experiments to investigate changes in mitochondrial ultrastructure (Fig 4A). Mitochondria from the subject (Fig 4A, panels b and c) were very narrow and had either no visible cristae, or cristae that were markedly reduced in length compared to control (Fig 4A, panel a). Suppression of SLC25A46 phenocopied these abnormalities in mitochondrial architecture (Fig 4A, panels d and e). In addition, we observed a small number of mitochondria with parallel stacks of cristae in siRNA-treated cells (Fig 4A, panel f). The changes in cristae architecture are quantified in Fig 4B. We did not observe the mitochondrial constriction sites reported in COS-7 cells by Abrams *et al* (2015), which they interpreted as fission intermediates.

Disruption of the MICOS complex results in detachment of membrane junctions and parallel stacks of cristae (Harner *et al*, 2011; Hoppins *et al*, 2011; van der Laan *et al*, 2012), similar to the phenotype we observed in a small proportion of control cells treated with the SLC25A46 siRNA. To investigate whether the altered cristae architecture in subject and siRNA-treated fibroblasts was the result of disruption of the MICOS complex, we evaluated the steady-state levels of two key MICOS components, MIC60 (mitofilin) and MIC19 (CHCHD3), by immunoblot analysis (Fig 4C). Both proteins were substantially reduced in subject fibroblasts: MIC60 to 31% and MIC19 to 9% of control. The disruption of the MICOS complex was phenocopied in control fibroblasts treated with an siRNA directed against SLC25A46 (MIC60 34% and MIC19 to 21% of control) and was exacerbated

in subject fibroblasts by further knockdown of the mutant SLC25A46 protein (MIC60 14% of control and MIC19 undetectable). To test whether loss of the MICOS components would affect the stability of SLC25A46, we depleted MIC60 and MIC19 using specific siRNAs. Depletion of MIC60 resulted in the complete disappearance of MIC19, and in cells treated with an siRNA against MIC19, only a very small amount of MIC60 was detectable, demonstrating their interdependence in the MICOS complex (Fig 4D). Immunoblot analysis showed no change in the level of SLC25A46 in MIC19 knockdown cells, but a substantial increase in cells in which both MIC60 and MIC19 were undetectable (Fig 4D). These results demonstrate that SLC25A46 is required for the stability of the MICOS complex. They further show that the level of SLC25A46 is very sensitive to the presence of MIC60. It is worth noting that previous studies demonstrated striking mitochondrial fission in HeLa cells depleted of MIC19 (Darshi *et al*, 2011; Ott *et al*, 2015), and it was concluded that mitochondria lacking MIC19 are fusion incompetent (Darshi *et al*, 2011). Our results show that this phenotype is completely suppressed by the loss of SLC25A46 function.

## SLC25A46 interacts with components of the mitochondrial fusion machinery and the MICOS complex

Given the established links between Ugo1 and mitochondrial fusion, combined with the above results suggesting functional interactions between SLC25A46 and the MICOS complex, we decided to test for physical interactions by immunoprecipitation experiments after chemical cross-linking (Table 1). All of these experiments were carried out using antibodies directed against the endogenous proteins, avoiding overexpression artifacts. SLC25A46 co-immunoprecipitated with OPA1, the profusion outer membrane proteins MFN1 and MFN2, analogous to what is found with Ugo1, SAMM50, and two components of the MICOS complex (MIC60, MIC19). OPA1 co-immunoprecipitated the same subset of mitochondrial proteins, and, in addition, other members of the MICOS complex, and several subunits of the TIM and TOM complexes (Table 1). Finally, immunoprecipitation of MIC60 brought down OPA1, the MICOS complex, and several subunits of the TIM and TOM complexes. These results strongly suggest that SLC25A46 is part of a larger interactome that integrates the major cristae organizing proteins with proteins of the outer mitochondrial membrane, consistent with a role in cristae maintenance.

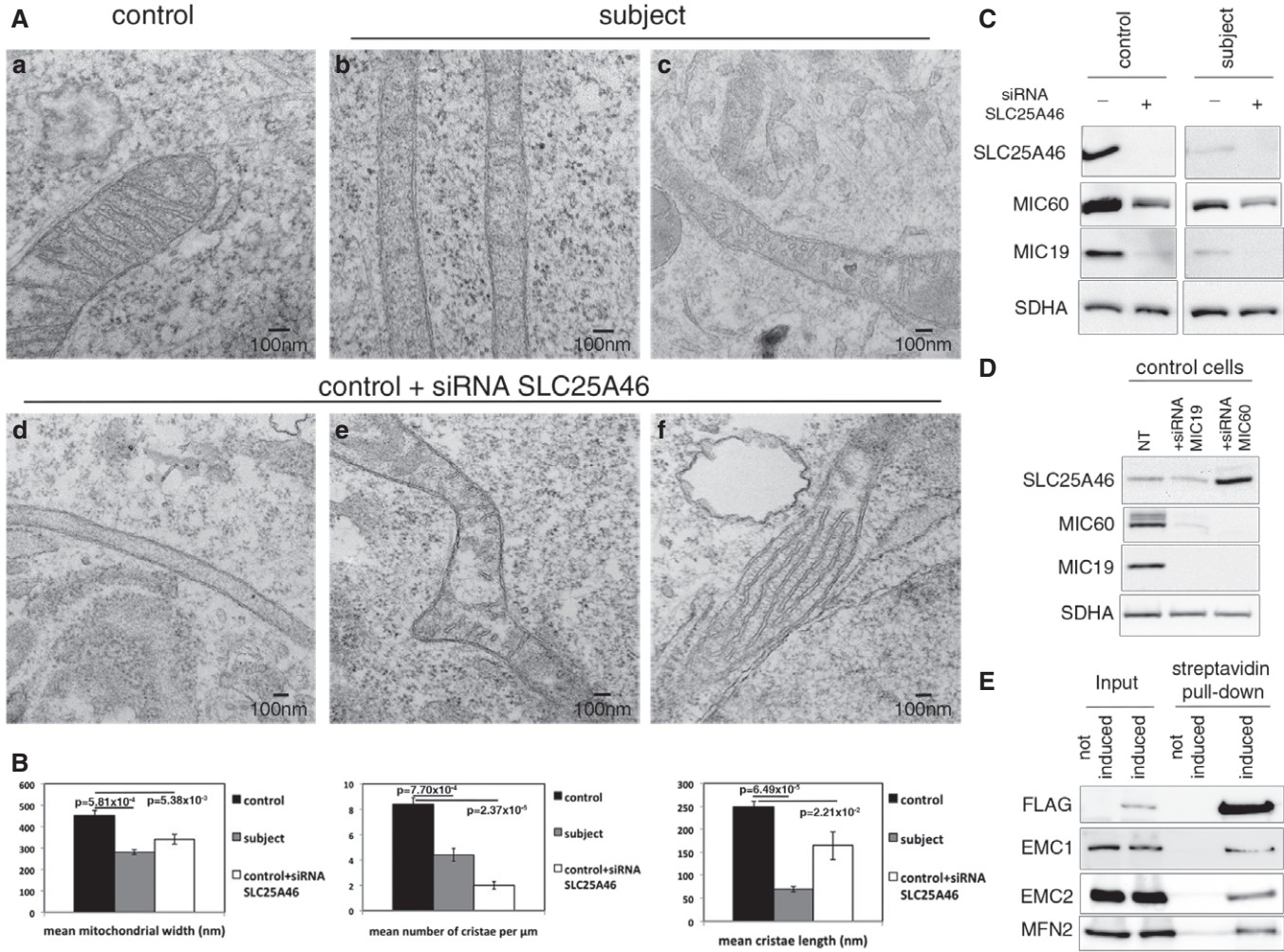

**Figure 4. SLC25A46 is required for mitochondrial cristae formation and interacts with the EMC complex in the ER.**

A   Analysis of the mitochondrial ultrastructure by transmission electron microscopy in control (a) and subject (b and c) fibroblasts and control fibroblasts treated with an siRNA against SLC25A46 (d–f).

B   Quantification of the effect of SLC25A46 on the mitochondrial ultrastructure organization. In each experimental condition, 30 cells were analyzed for mitochondrial width, the number of cristae per µm of mitochondrial length, and the length of cristae. Data represent mean ± SEM and P-values were calculated using a t-test.

C   Fibroblasts from control and subject, depleted of SLC25A46, were analyzed by immunoblot for expression of SLC25A46 and subunits of the MICOS complex. SDHA was used as a loading control.

D   Control fibroblasts, non-transfected or knocked down for MIC60 and MIC19, were analyzed by immunoblot for expression of SLC25A46, MIC60, and MIC19. SDHA was used as a loading control.

E   Immunoblot analysis of the BioID assay with an N-terminally FlagBirA*-tagged SLC25A46 showing an interaction of SLC25A46 with EMC1, EMC2, and MFN2.

Source data are available online for this figure.

## SLC25A46 interacts with components of the EMC complex, involved in lipid transfer from the ER to the mitochondria

The protein–protein interactions observed in the above experiments result from short-range molecular interactions that can be captured by chemical cross-linking. To further examine the interactome of SLC25A46, we used a complementary assay called BioID, or proximity-based biotinylation, a method enabling the detection of protein–protein interactions in living cells (Roux et al, 2012). SLC25A46 was N-terminally tagged with the promiscuous biotin-conjugating enzyme (BirA*) and expressed in the presence of biotin in HEK293 cells. This resulted in biotinylation of SLC25A46 neighboring proteins, which were then captured on streptavidin beads, and analyzed by mass spectrometry. This exploratory screen confirmed the interaction of SLC25A46 with the mitochondrial proteins reported above and, in addition, revealed an interaction with all of the components of an ER protein complex, the EMC (EMC1-9) (Appendix Table S1). The EMC was initially identified in a screen for proteins involved in protein folding in the ER (Jonikas et al, 2009) and was recently shown to be necessary for phospholipid transfer from the ER to mitochondria in yeast (Lahiri et al, 2014). Immunoblot analysis

**Table 1.  SLC25A46 interacts with mitochondrial fusion machinery, SAMM50 and the MICOS complex.**

| Co-immuno-precipitated proteins | Immunoprecipitated proteins (Mascot score of positive sample/control sample) | | | |
|---|---|---|---|---|
| | SLC25A46 | OPA1 | MFN2 | MIC60 (IMMT) |
| **Fission/fusion** | | | | |
| SLC25A46 | 363 | 281 | 50 | |
| OPA1 | 866 | 4,877 | 137 | 2,061 |
| MFN2 | 109 | 146 | 1,197 | |
| MFN1 | 70 | 301 | 208 | 155 |
| MFF | | 246 | | 115 |
| FIS1 | | 48 | | |
| MTFR1L | | 143 | | 166 |
| **MICOS complex** | | | | |
| MIC60 (IMMT) | 184 | 2,605/182 | | 3,638/106 |
| MIC19 (CHCHD3) | 142 | 866/107 | | 1,276/85 |
| MIC25 (CHCHD6) | | 287 | | 561 |
| MIC10 (MINOS1) | | | | 45 |
| MIC27 (APOOL) | | 111 | | 300 |
| APOO | | 64 | | 343 |
| DNAJC11 | | 589 | | 1,175 |
| **SAM** | | | | |
| SAMM50 | 88 | 980 | | 1,777 |
| **Porins** | | | | |
| VDAC1 | 309 | 1,412/164 | 112 | 1,061/68 |
| VDAC2 | 131 | 1,027/259 | 137 | 821/128 |
| VDAC3 | 53 | 864 | 68 | 385 |
| **Translocation complex** | | | | |
| TIMM8A | | 127 | | 142 |
| TIMM9 | | | | 217 |
| TIMM10 | | 76 | | 126 |
| TIMM13 | | 156 | | 148 |
| TIMM17B | | 54 | | 121 |
| TIMM23 | | 119 | | 111 |
| TIMM50 | | 587/22 | | 453 |
| TOMM7 | | 50 | | |
| TOMM20 | | 97 | | |
| TOMM40 | | 517 | | 562 |
| TOMM70A | 67 | 198 | | 377 |

Protein interactions were analyzed by co-immunoprecipitation after chemical cross- linking. SLC25A46, OPA1, MFN2, and MIC60 were immunoprecipitated with antibodies against the endogenous proteins and the co-immunoprecipitating proteins were identified by mass spectrometry. Mascot scores are shown (positive/control samples).

of the streptavidin–agarose pull-down from the BirA*-FLAG-SLC25A46 cells treated with biotin confirmed the interaction with EMC1 and EMC2, as well as the outer membrane protein MFN2 (Fig 4E). These data suggest that SCL25A46 may couple the lipid

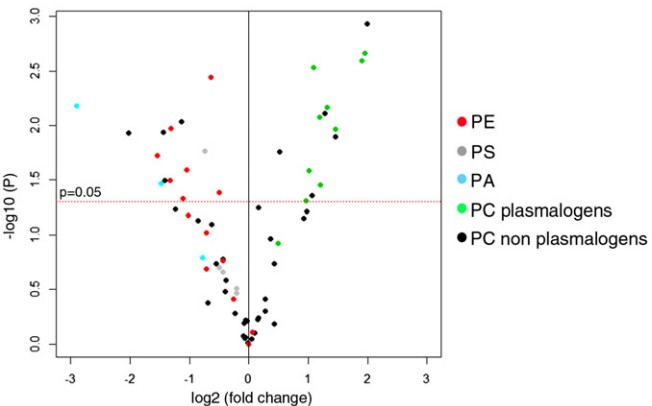

**Figure 5.  Loss of SLC25A46 leads to alterations in mitochondrial lipid content.**
Volcano plot representing the mass spectrometry signals from 71 glycerophospholipids, namely 49 PC (phosphatidylcholine) (39 non-plasmalogens and 10 plasmalogens), 14 PE (phosphatidylethanolamine), five PS (phosphatidylserine), and three PA (phosphatidic acid), which are shown as different color symbols as indicated, analyzed in gradient-purified mitochondrial extracts from subject vs. control cells. All lipid entities shown above the horizontal red line have a *P*-value < 0.05.

transport machinery from the ER to the regulators of mitochondrial architecture.

## Loss of SLC25A46 alters mitochondrial phospholipid composition and endoplasmic reticulum morphology

Phospholipid synthesis occurs largely in the ER, and mitochondrial membrane biogenesis and lipid metabolism require phospholipid transfer from the ER at regions of close contact between these organelles (Tamura *et al*, 2014). In addition to the acquisition of lipids for mitochondrial biogenesis, mitochondrial decarboxylase enzymes are essential for the conversion of phosphatidylserine (PS) to phosphatidylethanolamine (PE), which is returned to the ER for conversion to phosphatidylcholine (PC).

To test whether the altered mitochondrial network and mitochondrial architecture might result from a defect in mitochondrial lipid transfer and content, we carried out a targeted analysis of several glycerophospholipids in gradient-purified mitochondria from control and subject cells using mass spectrometry. Overall we were able to identify and quantify 71 lipid species (49 phosphatidylcholines (PC), 14 phosphatidylethanolamines (PE), five phosphatidylserines (PS), and three phosphatidic acids (PA) as shown in the volcano plot (Fig 5, Appendix Table S2). It is clear from the plot that most species of PE, PA, and PS are decreased in subject mitochondria. Interestingly, while PCs as a class did not show any specific pattern, all identified PC plasmalogens were increased in subject mitochondria.

Since PE from mitochondria is required to synthesize PC in the ER, we decided to investigate the potential consequences on ER morphology in subject cells. To test this, we examined ER morphology by immunofluorescence analyses (Fig 6). Using the anti-KDEL antibody, we observed a markedly abnormal ER morphology in subject cells, and in particular a loss of the tubular morphology in the periphery of the cell, and a more sheet-like structure overall (Fig 6A). We classified and quantified these phenotypes and

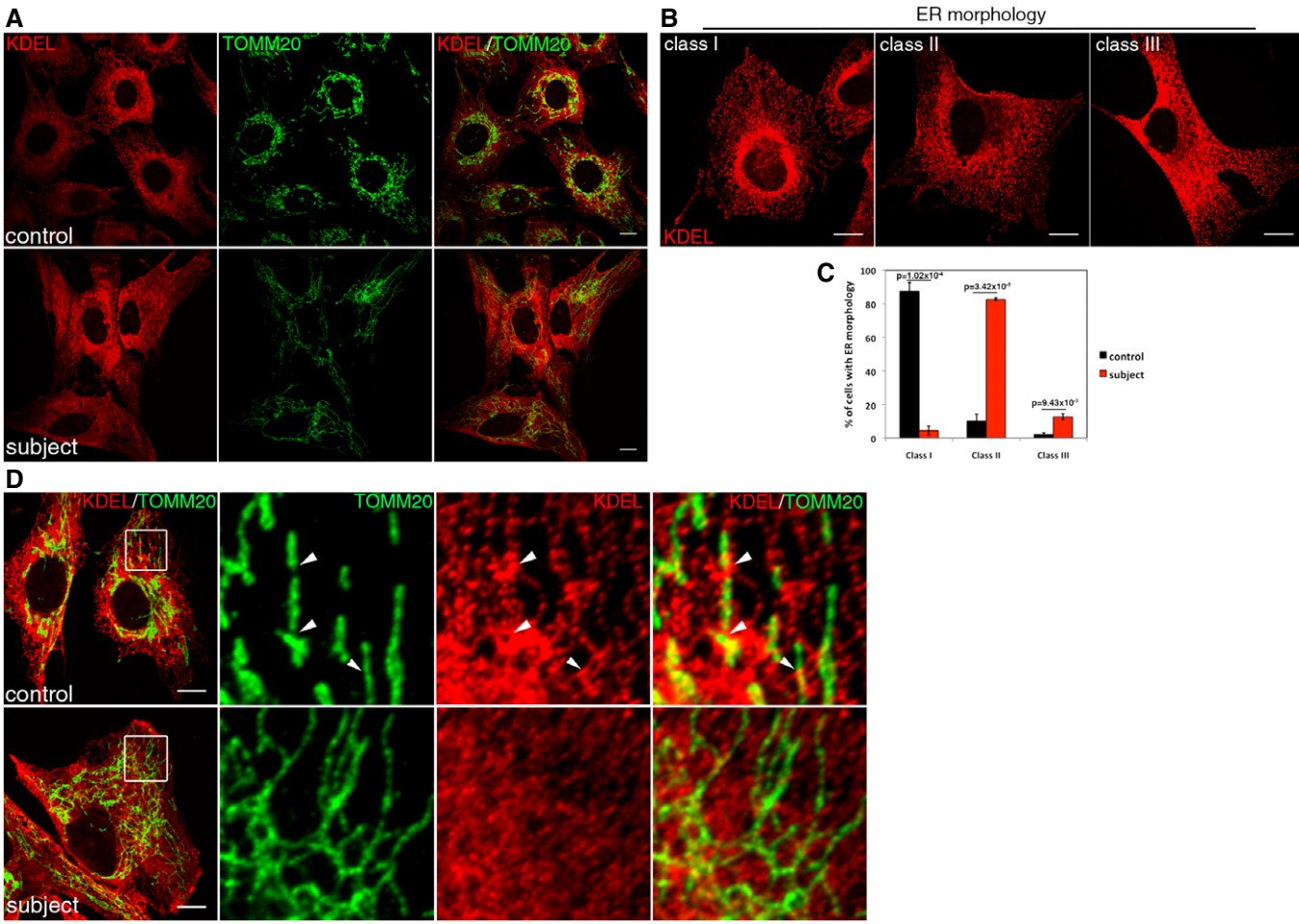

**Figure 6.   Loss of SLC25A46 alters endoplasmic reticulum morphology.**

A   Immunofluorescence analysis of the endoplasmic reticulum (ER) network in control and subject fibroblasts. The ER was stained in red (KDEL), mitochondria in green (TOMM20). Scale bars: 20 μm.

B   The organization of the reticular ER network was classified as class I (normal), class II (mildly disorganized), or class III (highly disorganized). Scale bars: 10 μm.

C   Quantification of the control and subject ER network organization. In each condition, 30 cells were analyzed in three independent experiments. Data represent mean ± SD and *P*-values were calculated using a *t*-test.

D   Immunofluorescence analysis of contacts sites between ER and mitochondria in control and subject fibroblasts. ER was stained in red (KDEL), mitochondria in green (TOMM20). Contact sites are shown by arrowheads. Scale bars: 10 μm.

showed an almost complete separation between subject and control cells (Fig 6B and C). While mitochondrial contacts were relatively easy to visualize in control cells, the sheet-like structure of the ER in subject cells obscured direct and unequivocal observation of such contacts (Fig 6D).

**Loss of SLC25A46 results in premature cellular senescence**

Although we immortalized the subject cells with retroviral vectors expressing hTERT and the E7 gene from HPV-16 (Lochmuller *et al*, 1999), we observed that their doubling rate was much reduced compared to controls (Fig 7A), or other fibroblast lines with OXPHOS assembly defects. This prompted us to measure markers associated with cellular senescence. The surface area of subject fibroblasts was twice that of controls (Fig 7B), and there was a fivefold increase in the proportion of cells expressing

senescence-associated β-galactosidase activity (Fig 7C and D). Using a cell migration assay, we also observed a marked decrease in cell migration (Fig 7F and G). At the molecular level, this was associated with an upregulation of p21 and altered processing of p53, concomitant with a downregulation of its regulator the E3 ubiquitin ligase P-Mdm2, all of which are associated with cell cycle arrest and senescence. Other molecular markers of senescence were also observed to increase, such as LAMP1, a lysosomal marker and the short forms of the intermediate filament protein vimentin. Retroviral expression of wild-type SLC25A46-GFP rescued these senescence and cell cycle phenotypes, with the exception of p21, which was only partially rescued (Fig 7E). We tested a several other cell lines for the senescence phenotype and observed that suppression of SLC25A46 suppressed the growth phenotype in all. Two examples are shown: a control primary fibroblast line (Fig 7H and I) and MCF7, a metastatic breast cancer line (Fig 7J and K).

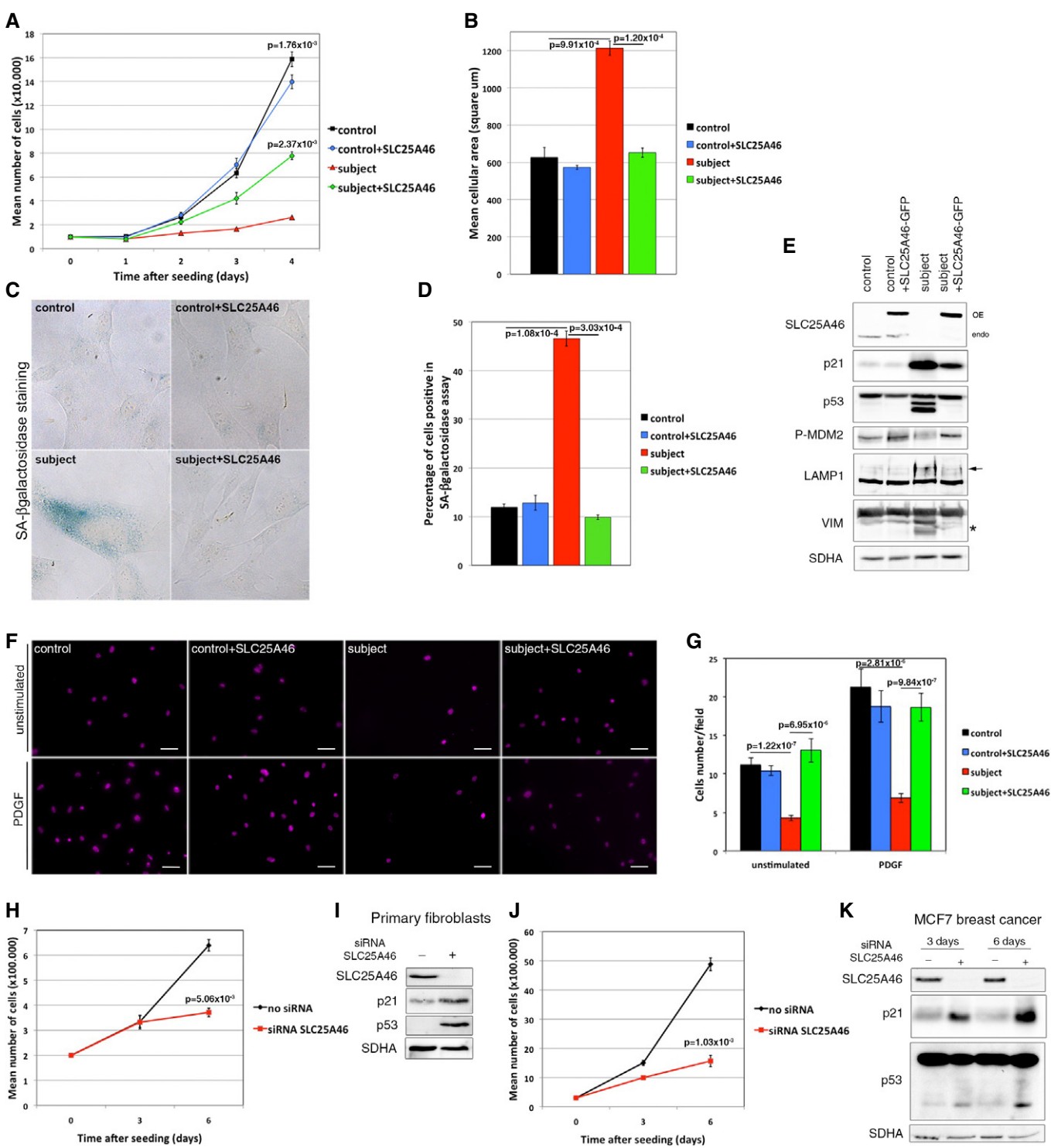

Figure 7.

## Discussion

This study identifies SLC25A46 as a key player in human mito-chondrial architecture, and, when mutated, as a cause of classical Leigh syndrome, an early-onset neurodegenerative disease. A recent study identified mutations in SLC25A46 in four families with optic atrophy, axonal CMT, and cerebellar atrophy (Abrams *et al*, 2015), and an additional patient was recently reported with an optic atrophy spectrum disorder (Nguyen *et al*, 2016), indicating that the neurological phenotypes produced by mutations in this gene are clinically heterogeneous. All reported cases of classical Leigh syndrome result from mutations in structural subunits of

**Figure 7.  Loss of SLC25A46 promotes premature senescence and impairs cell migration.**

A   Analysis of the cell growth rate of control and subject fibroblasts, overexpressing or not an EGFP-tagged version of the wild-type SLC25A46. Experiments were done in independent triplicates. Data represent mean ± SEM and P-values were calculated using t-test.

B   Analysis of the cell area in control and subject fibroblasts, overexpressing or not an EGFP-tagged version of the wild-type SLC25A46. 30 cells were measured in each condition and experiments were repeated three times. Data represent mean ± SEM and P-values were calculated using t-test.

C   Analysis of the senescence-associated β-galactosidase (SA-β-gal) activity in control and subject fibroblasts, overexpressing or not an EGFP-tagged version of the wild-type SLC25A46. Hydrolysis of X-gal results in the accumulation of a distinctive blue color in senescent cells.

D   Quantification of the senescence-associated β-galactosidase (SA-β-gal) activity in each of the experimental conditions described in (C). In each condition, 200 cells were analyzed. Percentage of positive blue cells corresponding to senescent cells was calculated. Experiments were done in independent triplicates. Data represent mean ± SEM and P-values were calculated using a t-test.

E   Immunoblot analysis from the same samples as in (A), for expression of SLC25A46, regulators of the cell cycle, and markers of senescence. SDHA was used as a loading control.

F   Analysis of cell migration capacity of control and subject fibroblasts, overexpressing or not an EGFP-tagged version of the wild-type SLC25A46, using a transwell assay as described in Materials and Methods. Cells that migrated through the membrane were identified under a microscope using DAPI staining. Scale Bars: 50 μm.

G   Quantification of cell migration capacity in each of the experimental conditions described in (F). In each condition, all cells present in a field were counted and five random fields were analyzed per condition. Experiments were done in independent triplicates. Data represent mean ± SD and P-values were calculated using a Student's t-test.

H   Analysis of the cell growth rate of primary control fibroblasts treated or not with an siRNA against SLC25A46 for 3 and 6 days. Experiments were done in independent triplicates. Data represent mean ± SEM and P-values were calculated using t-test.

I   Immunoblot analysis from the same samples as in (H), for expression of SLC25A46, regulators of the cell cycle, and markers of senescence. SDHA was used as a loading control.

J   Analysis of the cell growth rate of MCF7 breast cancer cells treated or not with an siRNA against SLC25A46 for 3 and 6 days. Experiments were done in independent triplicates. Data represent mean ± SEM and P-values were calculated using a t-test.

K   Immunoblot analysis from the same samples as in (J). SDHA was used as a loading control.

Source data are available online for this figure.

pyruvate dehydrogenase, structural subunits of the OXPHOS complexes, or in factors necessary for the assembly of the OXPHOS complexes. At least 75 different genes in nuclear and mitochondrial DNA genes have been shown to be causal (Ruhoy & Saneto, 2014; Lake et al, 2016), but to our knowledge this is the first report of a mutation in a protein influencing mitochondrial morphology that is associated with Leigh syndrome. Mutations in the fusion genes OPA1 and MFN2 have been associated with dominant optic atrophy (Alexander et al, 2000; Delettre et al, 2000) and CMT Type 2A (Zuchner et al, 2004), respectively, but the reasons for these genotype–phenotype correlations have remained an enduring mystery. Despite the striking abnormalities in mitochondrial architecture, the OXPHOS defect in the subject we studied, a specific decrease that resulted in complex IV assembly, was relatively mild. This, however, resulted in about a 50% decrease in basal cellular oxygen consumption, suggesting that coupled electron flow through the respiratory chain was additionally compromised. Our investigations on permeabilized fibroblasts, supplied with exogenous substrates, demonstrated that coupled electron flow was no different in the subject vs. control and that the maximum capacity of the respiratory chain (uncoupled electron flow) was unchanged in subject cells, indicating that substrate supply underlies the basal respiration deficit.

Both Ugo1 and SLC25A46 localize to the outer mitochondrial membrane and both physically interact with OPA1 (Mgm1 in yeast) and MFN2 (Fzo in yeast) [this study and Sesaki and Jensen (2004)]. Thus, both are poised to integrate the functions of the outer and inner mitochondrial fusion apparatus as has been proposed for Ugo1 (Hoppins et al, 2009). SLC25A46 also interacts with components of the MICOS complex, in particular the core components MIC60 and MIC19 (Table 1), and Ugo1 was reported to co-isolate with the components of the yeast MICOS complex (Harner et al, 2011). Despite these similarities, the loss of SLC25A46 results in a hyperfused mitochondrial

phenotype in human cells, whereas loss of Ugo1 function results in mitochondrial fission (Sesaki & Jensen, 2001, 2004; Hoppins et al, 2009).

While the exact molecular function of Ugo1 remains unknown, it is clear that in addition to its role in mitochondrial fusion, it is also important for normal mitochondrial architecture, as the number of cristae and mitochondrial contact sites (sites where the inner and outer membranes are closely apposed) is very dramatically reduced in Ugo1 deletion strains (Harner et al, 2011). We observed similar alterations in mitochondrial cristae architecture in subject fibroblasts and in cells depleted of SLC25A46. This failure to form normal cristae was associated with the cleavage of OPA1 to the short forms, and a marked decrease in the steady-state levels of MIC60, and the almost complete disappearance of MIC19. This suggests that the interaction with SLC25A46 is an important determinant of the stability of the MICOS complex. On the other hand, depletion of MIC19 had no effect on SLC25A46, while depletion of MIC60 resulted in a substantial increase in the steady-state level of SLC25A46, perhaps as an attempt to compensate for the loss of the MICOS complex, and in particular MIC60. Similar studies evaluating the effect of Ugo1 deletion on the MICOS components in yeast have not been reported, but would be of great interest. Studies implicating both OPA1 and the MICOS complex in the maintenance of cristae architecture are compelling, but how these machineries interact is unclear. The observation that SLC25A46 interacts with OPA1 and MIC60, the major MICOS organizer, provides a molecular link to integrate their function in modulating cristae architecture.

The marked alterations in mitochondrial morphology resulting from the loss of SLC25A46 are rather unexpected due to the clear role for yeast Ugo1 in facilitating fusion, although striking and similar abnormalities in cristae architecture from loss of function are common to both. Indeed, we observe an increase in OPA1 cleavage, which has also been shown to promote mitochondrial fragmentation

in dysfunctional mitochondria (Anand *et al*, 2014). We also show that the overall cellular content of DRP1 is increased, with efficient recruitment and oligomerization on mitochondria, in contrast to situations where the mitochondrial DRP1 receptors have been deleted (Osellame *et al*, 2016; Otera *et al*, 2016); however, it does not drive productive mitochondrial fission events. All of this might be explained if the primary function of SLC25A46 were to facilitate lipid flux at ER contact sites, rather than to coordinate fission *per se*. The decreased PS and PE content we observed in subject mitochondria is highly reminiscent of the changes observed in the mitochondrial phospholipid composition after deletion of five of the components of the EMC complex in yeast (Lahiri *et al*, 2014), which also results in a vegetative growth defect. Particularly striking in our results is the consistent increase in the PC plasmalogen content of subject mitochondria, the significance of which will require further investigation. Although the full function of the EMC complex is yet to be elucidated, it is interesting to note that mutations in EMC1 have recently been described in patients with a complex neurological syndrome that includes cerebellar atrophy (Harel *et al*, 2016).

The mitochondrial hyperfusion phenotype caused by the loss of SLC25A46 function culminates in premature cellular senescence. Fibroblasts from patients with OXPHOS deficiencies *per se* do not become senescent. While the deletion of Ugo1 results in an inability to grow on non-fermentable substrates, likely due to the loss of mtDNA (Sesaki & Jensen, 2004), cells can grow in rich glucose medium (Harner *et al*, 2011), although vegetative growth rate is reduced (Stevenson *et al*, 2001). Cellular senescence has also been described in mammalian cells depleted of DRP1 (Park *et al*, 2010) or FIS1 (Lee *et al*, 2007), both components of the mammalian mitochondrial fission machinery. Whether this reflects an inability to properly segregate mitochondria to the daughter cells during mitosis, or perhaps a role for mitochondrial dynamics in regulating senescence through signaling pathways, is not yet clear. Indeed, there are several reports of direct links between the GTPases that regulate morphology and signaling pathways including Notch (Kasahara *et al*, 2013), antiviral signaling (Castanier *et al*, 2010), and Ras (Chen *et al*, 2004).

It is reasonable to conclude that the hyperfused phenotype and lack of cristae in the subject cells are caused by a defect in lipid transfer between the ER and mitochondria. It has been established that mitochondrial fission occurs at sites of ER contact (Friedman *et al*, 2011); however, the molecular contributions of the ER in driving scission remain unclear. In cells lacking SLC25A46, changes in the lipid composition of the ER could inhibit membrane tubulation, which is critical to "wrap" the mitochondria at sites of division. Overall, our data are consistent with a model whereby lipid transfer from the ER may facilitate changes in mitochondrial curvature during division. In the absence of this, mitochondria remain hyperfused, even when OPA1 is in the short, fusion-incompetent form. Thus, our newly identified links between MFN2, OPA1, MICOS, and EMC indicate that SLC25A46 mechanistically couples lipid flux between the ER and mitochondria directly at outer/inner mitochondrial membrane contacts. This interactome also appears necessary to provide the flow of lipids essential for cristae growth and maintenance. Future experiments will better define the dynamic coordination between the complexes and functions that link to SLC25A46.

## Materials and Methods

### Human studies

The research studies on cell lines were approved by the institutional review board of the Montreal Neurological Institute, McGill University.

### Whole-exome and Sanger sequencing analysis

Whole-exome library preparation, capturing, and sequencing were performed at the high-throughput sequencing platform of the McGill University and Genome Quebec Innovation Centre, Montreal, Canada, as detailed in our previous publications (Fahiminiya *et al*, 2013, 2015; Hinttala *et al*, 2015). Briefly, genomic DNA was extracted from patient fibroblasts. Sequencing was performed on an Illumina Hiseq 2000 sequencer after the capturing of coding regions with the TruSeq Exome Enrichment Kit (Illumina, San Diego, CA). Using BWA (version 0.5.9) (Li *et al*, 2009), sequencing reads (100 bp paired end) were aligned to the human reference genome (UCSC hg19). Local realignment around small insertions or deletions (indels) and computation of depth of coverage were performed by Genome Analysis Toolkit (GATK, version 1.0.5) (McKenna *et al*, 2010). The average coverage of consensus coding sequence (CCDS) was 52X and 93% of bases were covered by $\geq$ 5 reads. Samtools (v. 0.1.17) (Li *et al*, 2009) mpileup and bcftools were used to call single nucleotide variants (SNVs) and indels. All called variants were subsequently annotated with in-house annotation pipeline that uses ANNOVAR and some custom scripts. The damaging effect of detected variants was predicted using three *in silico* prediction tools [SIFT (Kumar *et al*, 2009), PolyPhen-2 (Adzhubei *et al*, 2010), and MutationTaster (Schwarz *et al*, 2010)].

To identify genetic cause of the disease, we focused on exonic and canonical splice site variants with possible protein altering effects (nonsense, frameshift indel, splicing, and missense). We only considered variants that (i) covered by at least five reads, (ii) had a quality score higher than 20, and (iii) had an allele frequency < 5% in public databases (the 1,000 Genomes or in the Exome Variant Server, EVS) and seen in less than 20 individuals in our in-house exome database (contains ~1,000 previously sequenced samples) and not occurred as homozygotes in the ExAC database. Finally, the remaining variants were annotated with known and predicted mitochondrial proteins by using the MitoCarta database and were considered for further investigation.

Sanger sequencing of the *SLC25A46* gene in cDNA and gDNA from the subject was used to confirm the exome sequencing results. Total RNA was isolated from fibroblasts using the RNeasy kit (Qiagen, Toronto, ON. Canada) and the cDNA was amplified with specific primers using the one-step reverse transcriptase–polymerase chain reaction (RT–PCR) kit (Qiagen). Total genomic DNA was extracted from fibroblasts using the DNAeasy kit (Qiagen) and primers specific for exon 4 of *SLC25A46* were used to amplify the gDNA.

### Cell lines

Control and subject fibroblasts were immortalized as described previously (Lochmuller *et al*, 1999). MCF7 breast cancer cells and

fibroblasts were grown in high-glucose DMEM supplemented with 10% fetal bovine serum, at 37°C in an atmosphere of 5% $CO_2$. Cells stably overexpressing the HA-tagged or EGFP-tagged version of SLC25A46 were engineered using retroviral vectors as previously described (Weraarpachai *et al*, 2009).

### siRNA transfection

Stealth RNAi duplex constructs (Invitrogen) were used for transient knockdown of SLC25A46 in control and subject fibroblasts or MCF7 cells. Stealth siRNA duplexes were transiently transfected into cells using Lipofectamine RNAiMAX (Invitrogen), according to the manufacturer's specifications. The transfection was repeated on day 3 and the cells were harvested on day 6 for analysis.

### Mitochondrial isolation

Fibroblasts were rinsed twice, resuspended in ice-cold 250 mM sucrose/10 mM Tris–HCl (pH 7.4), and homogenized with ten passes of a pre-chilled, zero-clearance homogenizer (Kimble/Kontes). A post-nuclear supernatant was obtained by centrifugation of the samples twice for 10 min at $600 \times g$. Mitochondria were pelleted by centrifugation for 10 min at $10,000 \times g$ and washed once in the same buffer. Protein concentration was determined by Bradford assay.

### Cell fractionation

Fibroblasts were rinsed twice in PBS, resuspended in ice-cold 250 mM sucrose/10 mM Tris–HCl (pH 7.4). 10% were pelleted and lysed in 1.5% DDM in PBS. The lysate was centrifuged for 15 min at $20,000 \times g$, and the supernatant was considered as the total fraction. The rest of the cells were homogenized in ice-cold 250 mM sucrose/10 mM Tris–HCl (pH 7.4) with ten passes of a pre-chilled, zero-clearance homogenizer (Kimble/Kontes). A post-nuclear supernatant was obtained by centrifugation of the samples five times for 10 min at $600 \times g$. Mitochondria were pelleted by centrifugation for 10 min at $10,000 \times g$, washed twice in the same buffer, and lysed in 1.5% DDM in PBS. The lysate was considered as a crude mitochondrial fraction. The remaining supernatant, after the mitochondria fractionation step, was cleared by three further rounds of centrifugation at $10,000 \times g$ for 10 min. This supernatant was then centrifuged at $100,000 \times g$ for 1 h. The pellet was washed two times with PBS, lysed in 1.5% DDM in PBS, and considered as light membranes. The supernatant was adjusted at 1.5% DDM and considered as the cytosol fraction. Protein concentration was determined by Bradford assay.

### BMH cross-linking

Cross-linking experiments were performed as previously described (Prudent *et al*, 2015). Briefly, mitochondria were prepared in HIM buffer (200 mM mannitol, 70 mM sucrose, 1 mM EGTA, 10 mM HEPES [pH 7.5]) at 1 μg/μl. Half of the sample was kept without cross-linking, and the other half was treated with BMH (bis-maleimido-hexane) at 10 mM for 45 min at RT with rotation. The reaction was stopped by addition of DTT at 50 mM final for 15 min at RT with rotation. Mitochondria were then pelleted by centrifugation at $10,000 \times g$ for 10 min and rinsed twice with HIM buffer.

### Denaturating and native PAGE

For SDS–PAGE, cells were extracted with 1.5% DDM/PBS, after which 20 μg of protein was run on polyacrylamide gels. BN-PAGE was used to separate individual OXPHOS complexes. Isolated mitochondria were solubilized with 1% DDM, and 20 μg of solubilized samples were run in the first dimension on 6–15% polyacrylamide gradient gels as previously described (Leary, 2012). The separated proteins were transferred to a nitrocellulose membrane and immunoblot analysis was performed with the indicated antibodies.

### Oxygen consumption measurement

*SeaHorse*
Cells were seeded overnight in 24-well culture plate at $2 \times 10^5$ cells/well. Culture media were removed the following day and replaced with XF Assay Media from SeaHorse Bioscience (Billerica, MA) containing 4.5 g/l glucose and incubated at 37°C. Oxygen consumption rate (OCR) was measured using the SeaHorse XF24 Extracellular Analyzer over a 65-min time period. The assay consisted of two basal rate measurements followed by sequential injections of oligomycin (1.5 μM), FCCP (2 μM), and rotenone (0.1 μM) + antimycin (1.0 μM). Two rate measurements were performed after each injection. Experiments were done three times independently. Errors bars represent mean ± SEM and *P*-values were calculated using a *t*-test.

*Clark-type electrodes*
Respiration was measured polarographically using Clark-type electrodes (Hansatech Instruments, Norfolk, UK) (Burelle *et al*, 2015). Approximately $4 \times 10^6$ fibroblasts cells were permeabilized with digitonin (100 μg/ml) for 1 min at 37°C. Cells were then spun and resuspended in 1 ml of a respirometry buffer (300 mM Mannitol, 5 mM KCl, 0.5 mM EGTA, 5 mM $MgCl_2$, and 2 mg/ml BSA in 10 mM phosphate buffer; pH 7.55) (Rustin *et al*, 1994). Baseline respiration was recorded in the absence of respiratory substrates at 37°C followed by sequential addition of substrates: glutamate (5 mM) + malate (2.5 mM); ADP (2 mM); succinate (5 mM); oligomycin A (1 μM); CCCP (1 μM × 4 times) (control $n = 4$; subject $n = 5$). Complex IV-driven $O_2$ consumption was independently measured with the following sequence: antimycin A (4 μM); CCCP (3 μM); ascorbate (2 mM); TMPD/ascorbate (5 mM/20 mM); KCN (80 μM). Complex IV-driven $O_2$ consumption was defined as ($V_{O2}$ TMPD/ascorbate – $V_{O2}$ KCN) in order to correct for TMPD auto-oxidation (control $n = 7$; subject $n = 5$). Each measurement was normalized over protein content using the Bio-Rad DC protein assay kit, and results are expressed in nmole $O_2$/min/mg of protein. Error bars represent mean ± SEM and *P*-values were calculated using a *t*-test.

### Localization assays

To determine the membrane topology of SLC25A46, mitochondria were extracted with 100 mM alkaline carbonate at pH 11.5 as previously described (Weraarpachai *et al*, 2009) and the relevant fractions were analyzed by SDS–PAGE. To determine the localization of SLC25A46 at the membrane, 60 μg of freshly prepared crude mitochondria was put in the presence of an increasing concentration of

proteinase K diluted in the isolation buffer, for 20 min on ice. The reaction was stopped by adding PMSF (2 mM final) and the cells left on ice for 20 min. Mitochondria were then pelleted at 10,000 × *g* for 10 min, put in Laemmli buffer, and analyzed by SDS–PAGE.

### Immunofluorescence

Immunofluorescence analyses were performed as previously described (Prudent *et al*, 2015). Briefly, cells were fixed in 5% paraformaldehyde (PFA) in PBS at 37°C for 15 min, then washed three times with PBS, followed by quenching with 50 mM ammonium chloride in PBS. After three washes in PBS, cells were permeabilized in 0.1% Triton X-100 in PBS, followed by three washes in PBS. The cells were then blocked with 10% fetal bovine serum (FBS) in PBS, followed by incubation with primary antibodies in 5% FBS in PBS, for 1 h at RT. After three washes with 5% FBS in PBS, cells were incubated with the appropriate anti-species secondary antibodies coupled to Alexa fluorochromes (Invitrogen) (1:1,000) for 1 h at RT. After three washes in PBS, coverslips were mounted onto slides using fluorescence mounting medium (Dako).

### Confocal imaging

Stained cells were imaged using a 60× or a 100× objective lenses (NA1.4) on an Olympus IX81 inverted microscope with appropriate lasers using an Andor/Yokogawa spinning disk system (CSU-X), with a sCMOS camera.

Mitochondrial network morphology (Fig 2) was classified in a blinded manner as fused, intermediate, or fragmented. For each condition, 150 cells were analyzed. Experiments were done three times independently. Errors bars represent mean ± SEM and *P*-values were calculated using *t*-test.

To analyze mitochondrial DRP1 recruitment (Fig 3), a 0.2 μm *z*-axis image series of cells labeled for TOMM20 and DRP1 was obtained and stacked using the same conditions of gain, laser intensities, and exposure time. Images were then compiled as "max projection" and manually thresholded at the same level using control cells referenced with the FIJI software. Of note, due to the mitochondrial thinness and decreased fluorescence intensity, it was necessary to threshold mitochondria from subject samples manually for co-localization calculations. Mander's and Pearson's coefficients were determined between images in the two channels using the "coloc2" plug-in from FIJI.

Determination of the partitioning of DRP1 between the cytosol and mitochondria was analyzed as described previously (Palmer *et al*, 2013). Briefly, thresholded images were binarized, and total DRP1 fluorescence was measured. The binarized image from mitochondria was then subtracted from the total DRP1 signal, using the "Image calculator" plug-in from FIJI, resulting in a cytosolic DRP1 image. The fluorescence intensity of cytosolic DRP1 was then measured and subtracted from the total DRP1 fluorescence intensity to calculate the mitochondrial DRP1 signal. To create mitochondrial DRP1 "hot map", thresholded and binarized images from DRP1 and mitochondria were combined to create a mitochondrial DRP1 image using the "image calculator" plug-in from FIJI. This mitochondrial DRP1 created image was then merged with the mitochondrial image, and false color image showing the DRP1 accumulation at mitochondria was created. At least 15 cells per condition were analyzed for three independent experiments. Errors bars represent mean ± SD and *P*-values were calculated using *t*-test.

For ER morphology (Fig 6), three stacks of 0.2 μm were acquired for cells stained with anti-TOMM20 and anti-KDEL antibodies using the 60× objective. Images were then compiled by "max projection" using the FIJI software and quantified in class I, II, or III. At least 30 cells per condition were analyzed in three independent experiments. Error bars represent mean ± SD and *P*-values were calculated using a *t*-test.

### Analysis of the mitochondrial ultrastructure by transmission electron microscopy

Cells were washed in 0.1 M Na cacodylate washing buffer (Electron Microscopy Sciences) and fixed in 2.5% glutaraldehyde (Electron Microscopy Sciences) in 0.1 M Na cacodylate buffer overnight at 4°C. Cells were then washed three times in 0.1 M Na cacodylate washing buffer for a total of 1 h, incubated in 1% osmium tetroxide (Mecalab) for 1 h at 4°C, and washed with ddH2O three times for 10 min. Then, dehydration in a graded series of ethanol/deionized water solutions from 30 to 90% for 8 min each, and 100% twice for 10 min each, was performed. The cells were then infiltrated with a 1:1 and 3:1 Epon 812 (Mecalab):ethanol mixture, each for 30 min, followed by 100% Epon 812 for 1 h. Cells were embedded in the culture wells with new 100% Epon 812 and polymerized overnight in an oven at 60°C. Polymerized blocks were trimmed and 100 nm ultrathin sections were cut with an Ultracut E ultramicrotome (Reichert Jung) and transferred onto 200-mesh Cu grids (Electron Microscopy Sciences). Sections were post-stained for 8 min with 4% aqueous uranyl acetate (Electron Microscopy Sciences) and 5 min with Reynold's lead citrate (Fisher Scientific). Samples were imaged with a FEI Tecnai-12 transmission electron microscope (FEI Company) operating at an accelerating voltage of 120 kV equipped with an XR-80C AMT, 8 megapixel CCD camera. Based on those images, mitochondrial ultra structural characteristics (width, cristae/μm of mitochondria, length of cristae) were measured using ImageJ software. For each condition, 30 cells were analyzed. Error bars represent mean ± SEM and *P*-values were calculated using *t*-test.

### Immunoprecipitation

Before extraction, mitochondria (200 μg) isolated from control fibroblasts were chemically cross-linked with 1 mM disuccinimidyl glutarate (DSG) (Sigma) in isolation buffer, for 2 h on ice. The reaction was stopped by adding glycine pH 8.0 at 70 mM final for 10 min on ice. Mitochondria were pelleted, rinsed once, and extracted in 200 μl of lysis buffer (10 mM Tris–HCl pH 7.5, 150 mM NaCl, 1% n-dodecyl-D-maltoside (DDM) (Sigma), and complete protease inhibitors (Roche)) on ice for 30 min. The extract was centrifuged at 20,000 × *g* at 4°C for 20 min, and the supernatant was pre-cleared overnight with non-coated Dynabeads Protein A (Invitrogen) to reduce non-specific protein binding to the beads. Binding of indicated antibodies to Dynabeads Protein A (Invitrogen) was performed overnight. Antibodies were then cross-linked to the beads using 20 mM dimethyl pimelimidate (DMP) (Sigma). The immunoprecipitation reaction was performed overnight at 4°C. Samples were eluted using 0.1 M glycine pH 2.5/0.5% DDM, trichloroacetic acid precipitated, and analyzed by mass spectrometry

on an Orbitrap (Thermo Scientific) at the Institute de Recherches Cliniques de Montreal. The false discovery rate is < 5% with a Mascot score of 50.

## BioID

SLC25A46 was N-terminally tagged with BirA*-FLAG and integrated stably in HEK293 Flp-In T-REx (Invitrogen) cells for tetracycline inducible expression. Induction of expression, *in vivo* cellulose biotinylation and purification of biotinylated proteins followed by their identification by mass spectrometry, was performed as described previously, and data were compared to negative controls expressing the BirA* tag alone, the BirA* tag fused to GFP, and untransfected cells (Couzens *et al*, 2013). To validate the interaction of SLC25A46 with the EMC subunits, eluates from induced and non-induced cells were analyzed by immunoblot and the biotinylated partners were revealed with specific antibodies against EMC1, EMC2, MFN2, and FlagM2.

## Antibodies

Antibodies directed against the following were used in this study: SLC25A46 (Proteintech, 12277-1-AP), MFN2 (Santa Cruz, sc-100560), SDHA (Abcam, ab14715), AIFM1 (Chemicon, MABN92), ATP5A1 (Abcam, ab14748), HCCS (Proteintech, 15118-1-AP), SCO1 (in-house), MRPL14 (Proteintech, 15040-1-AP), NDUFA9 (Abcam, ab14713), UQCRC2 (Abcam, ab14745), COX4 (in-house), COX2 (in-house), MFN1 (Cell Signaling, 14739), IMMT/MIC60 (Abnova, H00010989-M01), CHCHD3/MIC19 (Aviva, ARP57040-P050), OPA1 (BD biosciences, 612607), MFF (Sigma, SAB1305258), CDKN1A/p21 (Cell Signaling, 2946), p53 (Abcam, ab176243), p-Mdm2 (Cell Signaling, 3521), LAMP1 (Santa Cruz, sc-18821), VIM (BD biosciences, 550513), VDAC1 (Abcam, ab14734), EMC1 (GeneTex, GTX119884), EMC2 (ProteinTech, 25443-1-AP) and FlagM2 (Sigma, F1804), TOMM20 (Santa Cruz, sc-11415), DRP1 (BD biosciences, 611113), p637-DRP1 (Cell Signaling, 6319), p616-DRP1 (Cell Signaling, 3455), KDEL (Abcam, ab50601), ITPR1 (Cell Signaling, 8568), alpha-tubulin (Santa Cruz, sc-23948).

## Cell growth assay

For immortalized fibroblasts, 10,000 cells were seeded in a six well culture dishes at day 0. After 1, 2, 3, and 4 days of culture, cells were trypsinized, homogenized, and counted using a Bio-Rad TC10 automated cell counter. Experiments were done in independent triplicates. Errors bars represent mean ± SEM and *P*-values were calculated using *t*-test.

For primary fibroblasts and MCF7 cells, 200,000 and 300,000 cells, respectively, were seeded in two 60-mm culture dishes at day 0 and submitted to siRNA. After 3 days of culture, a dish was counted, the second transfected for three more days. Experiments were done three times independently. Errors bars represent mean ± SEM and *P*-values were calculated using *t*-test.

## Cellular area measurement

Cells were seeded on coverglass, fixed, and mounted on a slide for observation. Images were captured using a Zeiss observer Z1

### The paper explained

#### Problem
Leigh syndrome is an early-onset, fatal, neurodegenerative disorder that is genetically heterogeneous. The genetic cause remains unknown in many Leigh syndrome patients and the mechanisms of pathogenesis are not well understood.

#### Results
Using whole-exome sequencing, we uncovered missense mutations in SLC25A46, a degenerate member of the mitochondrial metabolite carrier family. Unlike most members of this family, SLC25A46 localizes to the outer membrane. It interacts with components of the fusion/fission machinery (OPA1, MFN2), the complex that organizes mitochondrial cristae (MICOS), and the EMC complex in the ER, the yeast homologue of which has been proposed to act as a mitochondrial/ER tether. Mitochondria in subject fibroblasts were hyperfused, and there was a severe disruption of mitochondrial architecture. Most mitochondria were long and narrow with shortened or absent cristae. ER morphology was also substantially altered, becoming more sheet-like, in subject fibroblasts. These data suggested that SLC25A46 could be involved in facilitating lipid transfer between the ER and mitochondria, and lipidomic analyses of purified mitochondria from the subject showed a markedly altered phospholipid profile. Finally, loss of SLC25A46 function resulted in upregulation of markers of cellular senescence in mitotic cells in culture, suggesting that inefficient mitochondrial division, likely due to an insufficient supply of lipids, signals an exit from the cell cycle.

#### Impact
This is the first description of a protein involved in mitochondrial dynamics associated with Leigh syndrome. This case expands the clinical phenotype of mutations associated with mutations in SLC25A46, as most previously reported cases presented with a milder clinical phenotype. The newly identified links between MFN2, OPA1, MICOS, and EMC indicate that SLC25A46 mechanistically couples lipid flux between the ER and mitochondria directly at outer/inner mitochondrial membrane contacts, contributing to our understanding of the molecular pathology underlying Leigh syndrome.

inverted microscope, using a 40× Apochromat objective (Zeiss) and an ORCA-ER cooled CCD camera (Hamamatsu Photonics). Axiovision software (Zeiss) was used to acquire images and measure the cell surface. For each condition, 30 cells were measured, and the experiments were repeated three times. Errors bars represent mean ± SEM and *P*-values were calculated using a *t*-test.

## Senescence assay

Senescence-associated β-galactosidase (SA-β-gal) activity was determined using the Cellular Senescence Assay Kit (Millipore), according to the manufacturer's specifications. In this assay, SA-β-gal catalyzes the hydrolysis of X-gal, which results in the accumulation of a distinctive blue color in senescent cells. Cells were seeded in 6-well culture dishes at 50% confluency, and SA-β-gal activity was determined. For each condition, 200 cells were counted and evaluated for their blue color under a microscope. Images were captured using a Zeiss observer Z1 inverted microscope, with a 40× Apochromat objective (Zeiss) and an ORCA-ER cooled CCD camera (Hamamatsu Photonics). Axiovision software (Zeiss) was used to acquire images. Experiments were repeated three times independently.

Errors bars represent mean $\pm$ SEM and *P*-values were calculated using a *t*-test.

## Transwell migration assay

Cell migration was assessed using a transwell assay as described previously (Ferreira-da-Silva *et al*, 2015). Briefly, cells were serum-starved for 3 h at 37°C 5% $CO_2$ and resuspended in serum-free DMEM. The cells (10,000) were then seeded in the upper chamber of a 24-well transwell (Permeable Supports), using a 6.5-mm insert with 8.0-µm polycarbonate membrane (Costar), pre-coated with 10 µg/ml fibronectin from bovine plasma (Sigma). The lower chamber was filled with serum-free DMEM or serum-free DMEM supplemented with 30 ng/ml human platelet-derived growth factor (Millipore). Cells were incubated 16 h at 37°C 5% $CO_2$ and then fixed in 5% PFA. After three washes in PBS, non-migrating cells were removed. Filters were then removed and mounted on 10-well diagnostic microscope slides (Thermo Scientific), in Vectashield mounting medium containing DAPI (Vector Laboratories). Nuclei of cells that had migrated were visualized using DAPI staining on a Zeiss Imager M2 microscope equipped with a Plan Apochromat 20× 0.8 NA objective. Images were generated using Stereo Investigator software (MBF Bioscience). Each cell present in a field was counted and five random fields were analyzed per condition in three independent experiments. Error bars represent mean $\pm$ SD and *P*-values were calculated using a *t*-test.

## Lipidomic analysis using liquid chromatography–mass spectrometry (LC-MS)

Procedures for lipid extraction and analysis by LC-MS are essentially as described in references (Matyash *et al*, 2008; Sandra *et al*, 2010). Briefly, lipids were extracted from isolated mitochondria (100 µg protein), which had been spiked with six internal standards. Samples (equivalent to 1.3 µg of protein extract) were injected onto a 1290 Infinity HPLC coupled with a 6530 accurate mass QTOF (Agilent, Santa Clara, USA) via a dual electrospray ion source. Lipids were eluted onto a Zorbax Eclipse plus C18, 2.1 × 100 mm, 1.8 µm (Agilent, Santa Clara, USA) kept at 40°C with a gradient of 83 min and were analyzed in both negative and positive scan mode. Area signal was normalized using an internal standard of glycerophosphatidylglycerol (15:0/15:0). Tandem MS was performed to confirm lipids subclass of glycerophosphocholines (PC), glycerophosphoethanolamines (PE), glycerophosphoserines (PS), and glycerophosphates (PA). The experiment was repeated three times. Normalized data were analyzed using an unpaired *t*-test and statistical significance was set at $P < 0.05$.

## Statistical analysis

All data are reported as means $\pm$ SEM or means $\pm$ SD as indicated in the figure legend. Statistical significance was determined using Student's two-tailed unpaired *t*-tests. *P*-values < 0.05 were considered statistically significant and labeled as follows: *$P < 0.05$, **$P < 0.01$, and ***$P < 0.001$.

**Expanded View** for this article is available online.

## Acknowledgements

This research was supported by a grant from the CIHR to EAS (MT15460), CIHR 123398 to HMM, CIHR to CDR (9505), and an NSERC Discovery grant RGPIN-2014-06434 and Accelerator Supplement RGPAS 462169 to ACG. JP was supported by CIHR Postdoctoral Fellowship (MFE-140925). The Lunenfeld proteomics facility is partially supported by a Genome Canada Genomics Innovation Network (GIN) node grant, administered through the Ontario Genomics Institute. ACG holds a Canada Research Chair in Functional Proteomics and the Lea Reichmann Chair in Cancer Proteomics. We thank the Facility for Electron Microscopy Research (FEMR) of McGill University. We thank Tamiko Nishimura for fruitful discussions.

## Author contributions

AJ led the project, designed, and performed experiments and co-wrote the manuscript. JP performed fluorescence microscopy analysis of DRP1 localization and ER morphology and migration experiments. VP provided cell extracts knocked down for MIC60 and MIC19 and performed the oxygen consumption experiments using Seahorse. NS performed the oxygen consumption experiments using Clark-type electrodes. SF and JM designed and performed the genetics analysis. GM diagnosed and managed the subject. Z-YL and A-CG designed and performed the BioID analysis. CDR and AF supervised and carried out the lipid analyses. HMM contributed to experimental design. EAS helped with experimental design, supervised the project, and co-wrote the manuscript.

## Conflict of interest

The authors declare that they have no conflict of interest.

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
