## [Review Process File · EMBO Molecular Medicine]

SLC25A46 is required for mitochondrial lipid homeostasis and cristae maintenance and is responsible for Leigh syndrome

Alexandre Janer, Julien Prudent, Vincent Paupe, Somayyeh Fahiminiy, Jacek Majewski, Nicolas Sgarioto, Christine Des Rosiers, Anik Forest, Zhen-Yuan Lin, Anne-Claude Gingras, Grant Mitchell, Heidi M. McBride, Eric A. Shoubridge

Corresponding author: Eric A. Shoubridge, McGill University

Review timeline:

Submission date:	21 December 2015
Editorial Decision:	12 January 2016
Revision received:	21 April 2016
Editorial Decision:	19 May 2016
Revision received:	02 June 2016
Editorial Decision:	06 June 2016
Revision received:	07 June 2016
Accepted:	09 June 2016

Transaction Report:

Editor: Roberto Buccione

1st Editorial Decision

12 January 2016

Thank you for the submission of your manuscript to EMBO Molecular Medicine. We have now heard back from the three very expert Reviewers whom we asked to evaluate your manuscript.

As you will see, all three Reviewers highly praise the superb technical quality and rigor of your experimentation and recognise that the evidence supports the conclusions. However, Reviewers 1 and 2 note on one hand the somewhat compromised novelty and on the other that the mechanistic implications of your findings, which could have represented a significant advance with respect to the Nature Genetics paper, remain underdeveloped. I should add that these reservations reflect my very own when I made an initial editorial decision to send the manuscript out for peer-review. The reviewers also list a number of other issues.

In conclusion, while publication of the paper cannot be considered at this stage, given the potential interest of your findings and after internal discussion, we have decided to give you the opportunity to address the criticisms.

We are thus prepared to consider a substantially revised submission, with the understanding that the Reviewers' concerns must be addressed with additional experimental data where appropriate and that acceptance of the manuscript will entail a second round of review. The overall aim is to

significantly upgrade relevance and conclusiveness, especially with respect to the pathomechanisms. For instance, Reviewer 2 suggests some possible approaches in that respect and suggests provision of additional clinical characterisation.

I understand that if you do not have the required data available at least in part, to address the above, this might entail a significant amount of time, additional work and experimentation. I would therefore understand if you chose to rather seek publication elsewhere at this stage. Should you decide to do so, and we hope not, we would welcome a message to this effect.

Please note that it is EMBO Molecular Medicine policy to allow a single round of revision only and that, therefore, acceptance or rejection of the manuscript will depend on the completeness of your responses included in the next, final version of the manuscript.

EMBO Molecular Medicine now requires a complete author checklist (<http://embomolmed.embopress.org/authorguide#editorial3>) to be submitted with all revised manuscripts. Provision of the author checklist is mandatory at revision stage; The checklist is designed to enhance and standardize reporting of key information in research papers and to support reanalysis and repetition of experiments by the community. The list covers key information for figure panels and captions and focuses on statistics, the reporting of reagents, animal models and human subject-derived data, as well as guidance to optimise data accessibility. This checklist especially relevant in this case given the issues raised with respect to statistical treatment and animal numbers.

As you know, EMBO Molecular Medicine has a "scooping protection" policy, whereby similar findings that are published by others during review or revision are not a criterion for rejection. However, I do ask you to get in touch with us after three months if you have not completed your revision, to update us on the status. Please also contact us as soon as possible if similar work is published elsewhere.

I also suggest that you carefully adhere to our guidelines for publication in your next version, including our new requirements for supplemental data (see also below) to speed up the pre-acceptance process in case of a positive outcome.

***** Reviewer's comments *****

Referee #1 (Comments on Novelty/Model System):

There is only one model system. Immortalized human fibroblasts.

Referee #1 (Remarks):

The manuscript by Janer and colleagues investigates the role of SLC25A46 mutations as the cause of Leigh syndrome in a French Canadian consanguineous family. Whole exome- sequence identified a homozygote missense T142I mutation in SLC25A46. They show that SLC25A46 is drastically decreased in fibroblasts from the affected subjects. These cells had mild respiratory defects with complex IV decrease. They also had highly increased mitochondrial length, suggesting hyperfusion. It appears that Drp1 was recruited to the mitochondrial surface but that fission failed to be activated, despite an increase of Opa1 short isoforms. TEM detected profound changes in the morphology of mitochondria cristae, which were either absent or reduced in length or present in parallel stacks. Patient's fibroblasts had markedly reduced amounts of MICOS components. Using a combination of silencing and transgenic approaches, the authors convincingly demonstrated that the homozygote mutation in SLC25A46 caused protein destabilization, leading to MICOS disassembly and degradation. The results confirm that SLC25A46 is necessary for MICOS maintenance and normal cristae structure. Taken together the data further support the notion that SLC25A46 is a mammalian ortholog of yeast Ugo1. They also reaffirm that mutations in SLC25A46 cause mitochondrial syndromes. There are two novel aspects in the study, which have not been previously addressed by others. First is the interaction of SLC25A46 with EMC complex that was identified using a BioID strategy. These interactions may suggest that SLC25A46 is somehow involved in lipid transfer from the ER to mitochondria. Second is the increase of markers of cellular senescence observed in immortalized fibroblasts with the SLC25A46 mutation.

The work is overall very well done as far as the demonstration of the cellular and molecular effects of SLC25A46 mutations in fibroblasts derived from patients. All the appropriate controls, including siRNA and complementation approaches, have been utilized. The data relative to EMC are intriguing but quite preliminary, since no functional correlates are investigated to support the lipid transfer hypothesis. Similarly, the significance of the senescence marker expression in the immortalized cell lines from a single individual is limited.

Major points:

- 1) The paper by Abrahms et al. Nat Genet 2015 had identified mutations of SLC25A46 as responsible for mitochondrial neurological syndromes in a few families. In that study the similarity of SLC25A46 with UGO1 and its presumptive function as a component of the crista junctions and contact points in the context of MICOS interactions had been reported. Furthermore they had an in vivo model of the mutations. Therefore, this excellent work largely confirms the published observations.
- 2) The interactions with EMC are interesting, but there is no functional study supporting the hypothesis that mutations in SLC25A46 cause lipid transfer impairment and investigating the consequences.
- 3) The increase in the levels of SLC25A46 in MIC60 KD cells is counterintuitive and potentially very interesting. The mechanisms are not addressed.
- 4) The senescence phenotype should be investigated in primary (non transformed) cells and over multiple cell types, especially post-mitotic, with more direct disease relevance.
- 5) The decrease of complex IV activity does not explain why basal respiration is decreased by 50%, but maximal respiration is not. There must be some other regulation other than electron transfer that is impaired under basal (non-uncoupled) conditions, such as ATP/ADP translocation.
- 6) Hyper-fusion has been observed in fibroblasts and in other types of immortalized cells (Nat Genet 2015). The idea that despite Drp1 translocation fission fails because of loss of ER-mitochondrial interaction is intriguing, but further morphological and dynamics studies are needed to validate this hypothesis.

Referee #2 (Comments on Novelty/Model System):

The technical quality of the paper is excellent and the conclusions are clearly supported by evidence provided by a number of independent, converging results performed with high standard technical skills.

Since several mutations in the same gene have already been reported in a recent paper (Abrams et al, Nat Genet 2015), with a number of overlapping observations between the two papers, the originality of this contribution is limited, although several results, particularly those concerning the interactome of the mutant gene, are novel and interesting. The medical impact is relevant but not outstanding, because of the above considerations, and the fact that this paper reports a single case. It could be improved if some additional information were provided (for instance MRI images of the brain, details on the presence/absence of peripheral neuropathy and optic atrophy, similar to the patients reported by Abrams et al, cerebellar signs, etc.). In addition to the clinical aspects, the most relevant question raised by both papers is why the impairment of a gene encoding a protein arguably orthologue to a pro-fusion mitochondrial yeast factor, UGO1, is unexpectedly associated with a hyperfusion phenotype, suggesting an opposite role, i.e. pro-fission. However, neither Abrams et al nor Janer et al give a mechanistic contribution to explain the role of SLC25A46 and the functional divergence with its yeast orthologue (or paralogue) protein. I think an answer to these questions would be the really original contribution of the Janer et al. paper, sciencewise. In the present version, the paper is very rich in interesting findings and observations but inconclusive as far as the pathomechanism is concerned.

Referee #2 (Remarks):

This is an interesting, technically excellent, and medically relevant, paper, although its originality and novelty are somehow blunted by the results of an already published paper (Abrams et al, 2015), in which several mutations of the same gene and similar experimental observations are reported,

including the striking hyperfusion phenotype in mutant cells, the phylogenetic similarity of SLC25A46 with UGO1, some ultrastructural abnormalities, etc. However, Janer et al provide additional, relevant and convincing information about the interactors of the SLC25A46 gene product, in particular the physical and possibly functional connection with a number of proteins involved in mitodynamics and cristae architecture. Some points have to be considered:

- According to Abrams et al, the SLC25A46 cDNA does not rescue the phenotype of the delta-UGO1 strain and in fact the ablation of the former gives an opposite phenotype relative to the ablation of the latter. Therefore, there is little ground to consider the two proteins as orthologues, unless an experimentally proven hypothesis is proposed to explain these results. An alternative possibility is that the two gene products have diverged in their function, and their genes should therefore be considered as paralogous rather than orthologues.

- Abrams et al do not show specific activities of single respiratory chain complexes, therefore the finding of isolated COX defect is an original observation of Janer et al, but do the Authors have any hypothesis to explain such a specific defect? Is this reproducible by, for instance, siRNA of the gene in control cells? In addition, do the Authors have an explanation for the reduction of basal oxygen consumption, but normal maximal oxygen consumption by the Seahorse assay? Notably, this result is different from that shown by Abrams et al (fig. 4C).

- Clinical features: Leigh disease is an essentially neuropathological (or neuroimaging) entity characterized by symmetrical necrotic lesions through the brainstem and up to the basal ganglia. On the other hand, the phenotype(s) reported by Abrams et al include(s) optic atrophy, CMT-like peripheral neuropathy, but also cerebellar atrophy with symmetric lesions in the white matter surrounding both dentate nuclei, and severe atrophy of the cerebral cortex, therefore a complex syndrome combining peripheral abnormalities and lesions of the central nervous system (including optic atrophy). It would be interesting to have more details, if available, about investigations on fundus oculi/optic nerve, peripheral nervous system physiology, and brain MRI (in a supplementary figure), showing the main features concordant with the definition of Leigh disease.

- In addition to the clinical aspects, the most relevant question raised by both papers is why the impairment of a gene encoding a protein arguably orthologue to a pro-fusion mitochondrial yeast factor, UGO1, is unexpectedly associated with a hyperfusion phenotype, suggesting an opposite role, i.e. pro-fission. The work by Janer et al suggests physical interaction with pro-fusion factors, in addition to an ER complex involved in lipid exchange with mitochondria. Could the SLC25A46 protein be an inhibitor of pro-fusion factors? As recently suggested by Anand et al, OPA1 isoforms can determine opposite effects, i.e. pro-fusion or pro-fission, therefore investigation on the OPA1 isoform pattern in mutant or siRNA cells could give some useful hints. It would also be interesting to evaluate the effect on mitochondrial network, OPA1 isoforms, MFN1 and 2 amount, and DRP1 distribution in cells overexpressing SLC25A46.

In the present version, the paper is very rich in interesting findings and observations but rather inconclusive in elucidating the pathomechanism of the disease and the function of the protein.

Referee #3 (Remarks):

The authors describe a patient with Leigh syndrome, a fatal encephalopathy of infancy, who harboured a homozygous mutation in the nuclear-encoded mitochondrial gene SLC25A46, which they demonstrate is the human ortholog of yeast Ugo1p, a mitochondrial outer membrane-localized fusion factor. Using a set of biochemical and genetic approaches, the authors provide strong evidence that SLC25A46 is indeed the etiologic protein. Notably, the respiratory chain deficits are relatively "mild," except for complex IV. Mechanistically, they show that SLC25A46 plays a role in mitodynamics, that it interacts with the MICOS complex, and that it may be involved in lipid homeostasis.

The finding that SLC25A46 is the human homolog of Ugo1 confirms, and notably extends upon, the recent work of Abrams et al. Coupled with the excellent quality of the work performed, this paper is an important advance, in both the areas of mitochondrial disease (e.g. this is the first documented case of LS that is fundamentally divorced from deficits in "primary" OxPhos genes) and the basic biology of mitochondria. I have only minor comments.

While Ugo1 and SLC25A46 appear to be evolutionarily-related homologues, the functional similarity of the two proteins is somewhat vitiated by the fact that Ugo1 is inserted into the MOM by a non-canonical pathway involving Mim1 (Papic et al., referenced by the authors), which currently has no identified mammalian homologue. Is there any evidence that SLC25A46 insertion is similar?

The specific reduction in complex IV, coupled with a potential role for SLC25A46 in phospholipid transfer, implies a potential connection to cardiolipin (although CL is also important for complex III function, especially in supercomplexes). Any thoughts on this?

In Fig. 2A, wt SLC25A46 is a carbonate-resistant outer membrane protein. What about the mutant? If the mutation prevents membrane insertion, it might help explain the phenotypes you observe.

In Fig. S1, given the non-canonical nature of SLC25A4, it might be useful to add MTCH1 and MTCH2 to the alignments, if indeed such alignments are possible.

In Fig. S2, it was extremely difficult for me to see what was going on in the context of what the authors were trying to show. On my paper printout (but admittedly less so in the pdf itself) it was hard to tell if the blue mitochondria (and even the blue TOM20 label was hard to see) are more elongated in the patient, and whether DRP1 punctae are localized predominantly to mito tips (even in the control). It appears as if there are DRP1 punctae along the fused mitochondria in both control and patient cells (not unexpected), but with more "intra-mitochondrial" punctae in the patient (unexpected?), but without quantitation it is hard to know if this is the case, or even if DRP1 is relevant to the hyperfusion phenotype, which I assume is one of the points of the figure. Please fix or make clearer in some way (e.g. false coloring?)

1st Revision - authors' response

21 April 2016

We thank the reviewers for their insightful and positive comments on our manuscript. We believe that we have addressed the major questions relating to the molecular mechanism of action of SLC25A46. We now provide strong evidence demonstrating that the mitochondrial hyper fusion phenotype does not result from a failure to recruit or oligomerize DRP1 at the mitochondria. In addition there is an increase in the pro-fission short forms of OPA1. Thus the hyper fused phenotype cannot be explained by a lack of the pro-fission molecular machinery. We demonstrate a significant alteration in the phospholipid content in gradient-purified mitochondria from the subject, the pattern of which is similar to that described in the budding yeast when subunits of the EMC complex in the ER are deleted. We show, in addition, a very significant alteration in ER morphology, consistent with disrupted lipid transfer between mitochondria and the ER. Below we offer a point-by-point response to their queries.

Referee # 1.

- 1) We agree that the paper by Abrahms et al was excellent, but it is not our impression that they investigated the molecular mechanism of action of SLC25A46. Indeed they did not report any effects on MICOS stability, nor did they provide any detailed analysis of the ultra structural abnormalities in mitochondria, both of which we investigate in the present manuscript. They did produce a very nice vivo model in zebra fish, but as far as we can tell, this did not help in elaborating the molecular function of SLC25A46.
- 2) We agree entirely with the reviewer and now provide data from a lipidomics study, in which we identified and quantitated 72 different species of glycerophospholipids, clearly demonstrating an altered phospholipid composition in mitochondria from the subject (Figure 5). Significantly, the alterations in phospholipid content are very similar to those that have been reported in the budding yeast on disruption of the EMC complex (ref).
- 3) We agree that this is an interesting observation, and it perhaps represents some form of compensation, but this is not the focus of this manuscript, and we do not have a mechanism.

- 4) We have now investigated the senescence phenotype in a number of cell lines including primary non-immortalized human fibroblasts and a variety of tumour cell lines. We observed decreased doubling rates in all cells examined on suppression of SLC25A46, associated with upregulation of molecular markers of senescence. We show examples from the primary fibroblast cell line and from an aggressive breast cancer cell line (MCF7) in Figure. 7. We do not have post-mitotic cells available for this analysis.
- 5) We have now provided additional analyses of respiration in subject cells by examining oxygen consumption in digitonin-permeabilized fibroblasts using a conventional oxygen electrode (Fig. 1F). We show that basal oxygen consumption is indeed reduced (similar to results obtained on the Seahorse without addition of mitochondrial substrates), but there is in fact no difference between the subject and control when substrates (glutamate/malate, succinate) are added. Further, we show no difference in maximal uncoupled electron flow (on addition of CCCP) between the subject and control, in agreement with the results obtained on the Seahorse. We note that these rates are considerably higher relative to the basal rates using the conventional oxygen electrode compared to what is measured by Seahorse. In fact our experience is that the Seahorse consistently under estimates maximum electron flow. All of this suggests a problem with substrate delivery in the deficiency in basal oxygen consumption, which we mention in the text.
- 6) We have now carried out an extensive analysis of DRP1 recruitment to mitochondria, of ER morphology and, mitochondrial-ER contacts. (Figures 3 and 6). We show that steady-state levels of DRP1 are increased in subject cells, that it is efficiently recruited to mitochondria, it oligomerizes, its phosphorylation state (at Ser 616, 637) is unaltered compared to control (Supplementary figure 2), yet it does not appear to promote scission events. We also show that ER morphology is disturbed in subject cells, becoming much more sheet-like with loss of tubulation. This phenotype makes it difficult to unequivocally identify points of ER-mitochondrial contact; however, it strongly suggests an altered interaction between mitochondria and the ER.

Referee #2.

- 1) We thank this reviewer for the positive comments. We have now compared the clinical course of our patient with that of previously-reported patients, which were generally much milder. We fully agree that inclusion of MRI and fundus images would have enhanced the description. Unfortunately, in preparing the article we discovered that all radiological images on X-ray films in our hospital's archives were destroyed. Therefore, regrettably, no images are available. However the most important features of the MRI and the neuro-ophthalmologic examination are now described in more detail than previously (in the Supplemental material). Comparison with reported cases shows that our subject overlaps in some ways with the clinical presentation of reported cases (mild paleness of the optic disks, basal ganglion involvement, hyperactive deep tendon reflexes, and in one previously-reported patient, progressive atrophy of the hindbrain). The subject also extends the known clinical spectrum of *SLC25A46* deficiency to include Leigh disease and infantile death. We now discuss these points after the clinical description in the Supplemental material. Again we thank the reviewer because this modification provides perspective and improves the quality of the article for clinical readers.
- 2) We now provide a plausible molecular mechanism that can explain the severe disruption of mitochondrial architecture in both mammals and yeast that are seen when SLC25A46/Ugo1 function is suppressed or lost. We hasten to add that this important feature of Ugo1 loss of function has not emphasized in previous reports that only focused on mitochondrial fusion.
- 3) We respectfully disagree with this reviewer's discussion of orthologues vs. paralogues. Genes are orthologous if derived from a common ancestor (without reference to their present day function). The extensive phylogenetic analysis performed in (Haferkamp and Schmitz-Esser, 2012) clearly shows that SLC25A46 and Ugo1 form a separate branch on the phylogenetic tree, and this is a reason to consider them orthologous. Paralogues are gene duplications that co-occur in the same organism, having arisen from a gene duplication event; there is nor evidence that paralogues of either protein exist in yeast or mammals.
- 4) We do not know why we only see a rather small defect in complex IV assembly and we did not examine this in siRNA experiments in control cells. We now have done further extensive experiments exploring the oxygen consumption defect (see answer 5 above). Consistent with the assembly defect, we see a small decrease in maximum catalytic activity of complex IV measured with TMPD/ascorbate (Fig. 1F)

- 5) We have added additional clinical details as outlined in response 1 above. Unfortunately we are not able to supply the original MRI as the film was destroyed.
- 6) Indeed we investigated the OPA1 isoform pattern in subject and siRNA cell lines and show an accumulation of the short (pro-fission) isoforms (Figure 2C), and we have now done an extensive analysis of DRP1 recruitment to mitochondria as outlined in the response above.
- 7) We now provide evidence of a disturbance in the phospholipid composition of mitochondrial membranes, providing a pathomechanism for the phenotypes we observed in subject and siRNA cells.

Referee #3

- 1) We have not investigated the insertion of SLC25A46 into the OM, which is indeed an interesting biological question. However, the protein in the patient is very hard to detect so these studies would be difficult to pursue, and are not really the focus of the paper, which is to elucidate the molecular pathology due to loss of SLC25A46 function. The amino acid substitution in the subject apparently results in an unstable protein, and this could be due to inefficient insertion in the membrane.
- 2) We do not have an explanation for the fact that only a mild defect in complex IV is seen in the subject cells. We show that the phospholipid composition of the mitochondrial membranes is altered, but exactly how that specifically affects complex IV assembly is not yet clear. We could not quantify CL in our mass spectrometry analysis.
- 3) We did not examine the mutant, but given that it has 6 transmembrane domains it is very unlikely that it could exist as a soluble mitochondrial protein, and it does enrich with isolated mitochondria, so it is not cytosolic (Figure 3C).
- 4) We have now added the MTCH1 and MTCH2 to the alignments. Thank you for that suggestion.
- 5) We completely agree with the reviewer. We have eliminated this figure and now show a detailed analysis of DRP1 recruitment (Fig. 3).

2nd Editorial Decision

19 May 2016

Thank you for the submission of your revised manuscript to EMBO Molecular Medicine. We have now received the enclosed reports from the referees that were asked to re-assess it. As you will see the reviewers are now globally supportive and I am pleased to inform you that we will be able to accept your manuscript pending the following final amendments:

- 1) While performing our pre-publishing quality control and image screening routines, we noticed that the 1st, 3rd and 4th panels of the GFP row in Fig 2D appear actually empty. I do not refer to the lack of GFP signal, which is to be expected based on the experimental condition, but to the total absence of any background signal whatsoever. Please provide an explanation and if the case, a modified figure.
- 2) We also noticed, during the above mentioned screen, excessive contrasting in Fig. 1 and Fig. EV2. Please provide better images. In this respect, provision of source data would help (see point 6 below).
- 3) Please provide your supplemental information in a single "appendix" PDF file as per our author guidelines (<http://embomolmed.embopress.org/authorguide#expandedview>). Please make sure you adjust the callouts accordingly in the manuscript.
- 4) Please incorporate the "The Paper Explained" section into the main manuscript file (it is currently a separate document).
- 5) As per our Author Guidelines, the description of all reported data that includes statistical testing must state the name of the statistical test used to generate error bars and P values, the number (n) of independent experiments underlying each data point (not replicate measures of one sample), and the

actual P value for each test (not merely 'significant' or ' $P < 0.05$ ').

6) We encourage the publication of source data, particularly for electrophoretic gels and blots, with the aim of making primary data more accessible and transparent to the reader. Would you be willing to provide a PDF file per figure that contains the original, uncropped and unprocessed scans of all or at least the key gels used in the manuscript? The PDF files should be labeled with the appropriate figure/panel number, and should have molecular weight markers; further annotation may be useful but is not essential. The PDF files will be published online with the article as supplementary "Source Data" files. If you have any questions regarding this just contact me.

Please submit your revised manuscript within two weeks. I look forward to seeing a revised form of your manuscript as soon as possible.

***** Reviewer's comments *****

Referee #1 (Comments on Novelty/Model System):

The revised manuscript has introduced new results on phospholipids alterations in the mutant cells. It has also provided additional information on the various cell types that extend the original results obtained in immortalized fibroblasts. The novelty issue on the homology with UGO1 and the role in disease is balanced by the careful and convincing experiments that investigate the functional role of SLC25A46.

Referee #2 (Comments on Novelty/Model System):

The paper has improved in the technical quality and quality of results. The analysis of the lipidome carried out by the Authors suggests a possible mechanism to solve the discrepancy between the effect of UGO1 ablation in yeast and the (opposite) effect of mutations in SLC25A46 in humans. Although this remains a speculative argument, the scientific information provided by this paper is relevant and warrants interesting and challenging work for the future.

Referee #3 (Remarks):

No comments.

2nd Revision - authors' response

02 June 2016

Thank you for the provisional acceptance of our manuscript EMM-2015-06159-V2.

Answers to your queries appear below.

- (1) Yes these were empty panels, and we should have indicated this. We have now removed them and left empty spaces. We did not think it useful to provide a picture of the background noise, which would simply appear black.
- (2) We have now provided better images, and the source data files for all gels and blots.
- (3) We have provided the supplemental data as a single PDF
- (4) We have incorporated the paper explained into the main manuscript.
- (5) We have now stated the statistical test used and the significance level in all instances.
- (6) We have provided the source data for the major figures with gels/blots.

I hope that you will now find the paper suitable for publication in EMM.

Corresponding Author Name:
Journal Submitted to:
Manuscript Number: